# N-cadherin directs the collective Schwann cell migration required for nerve regeneration through Slit2/3-mediated contact inhibition of locomotion

Julian JA Hoving[†], Elizabeth Harford-Wright[†], Patrick Wingfield-Digby, Anne-Laure Cattin, Mariana Campana, Alex Power, Toby Morgan, Erica Torchiaro, Victor Quereda, Alison C Lloyd*

UCL Laboratory for Molecular Cell Biology and the UCL Cancer Institute, University College London, London, United Kingdom

*For correspondence:
alison.lloyd@ucl.ac.uk

†These authors contributed equally to this work

Competing interest: The authors declare that no competing interests exist.

**Abstract** Collective cell migration is fundamental for the development of organisms and in the adult for tissue regeneration and in pathological conditions such as cancer. Migration as a coherent group requires the maintenance of cell–cell interactions, while contact inhibition of locomotion (CIL), a local repulsive force, can propel the group forward. Here we show that the cell–cell interaction molecule, N-cadherin, regulates both adhesion and repulsion processes during Schwann cell (SC) collective migration, which is required for peripheral nerve regeneration. However, distinct from its role in cell–cell adhesion, the repulsion process is independent of N-cadherin *trans*-homodimerisation and the associated adherens junction complex. Rather, the extracellular domain of N-cadherin is required to present the repulsive Slit2/Slit3 signal at the cell surface. Inhibiting Slit2/Slit3 signalling inhibits CIL and subsequently collective SC migration, resulting in adherent, nonmigratory cell clusters. Moreover, analysis of ex vivo explants from mice following sciatic nerve injury showed that inhibition of Slit2 decreased SC collective migration and increased clustering of SCs within the nerve bridge. These findings provide insight into how opposing signals can mediate collective cell migration and how CIL pathways are promising targets for inhibiting pathological cell migration.

## Editor's evaluation

Contact inhibition of locomotion (CIL) describes a process where collision between migrating cells induces a change in direction of cell migration leading to cell dispersion and Schwann cell migration has been shown to be essential for nerve repair. In this study, Hoving and colleagues demonstrate that CIL is an important driver of collective Schwann cell migration by using live cell imaging, cell migration assays and an ex vivo model of nerve injury and repair. They show that Slit2/3/Robo signaling is required to induce local cell repulsion between colliding Schwann cells, in an N-cadherin-dependent manner and further demonstrate the importance of an N-cadherin-glypican-4-Slit signaling axis during CIL. This study convincingly describes a new and important molecular mechanism of CIL in adult tissues and in the context of wound repair and is likely to be of interest to a broad range of biologists given the importance of both CIL and N-cadherin to a number of cellular and developmental processes.

## Introduction

Tissue and organ morphogenesis requires the orchestration of the movement of large numbers of cells (*Klämbt, 2009*; *Scarpa and Mayor, 2016*). In the adult, cell migration is less frequent but is important for aspects of tissue renewal and immune surveillance and can be activated following an

injury to contribute to wound healing and tissue regeneration (*Friedl and Gilmour, 2009*; *Moreau et al., 2018*; *Shaw and Martin, 2016*; *Worbs et al., 2017*). Moreover, these modes of migration are frequently recapitulated during pathologies such as cancer, allowing tumour cells to spread from their original site (*Friedl and Gilmour, 2009*; *Madsen and Sahai, 2010*; *Reymond et al., 2013*).

Peripheral nerve is one of the few tissues in the mammalian adult that retains the ability to regenerate following an injury (*Jessen et al., 2015*; *Poss, 2010*; *Zochodne, 2012*). We have previously shown that the successful regeneration of a transected nerve requires the collective migration of cords of Schwann cells (SCs) that transport re-growing axons across the injury site (*Parrinello et al., 2010*; *Stierli et al., 2019*; *Stierli et al., 2018*). Moreover, SC cords gain directionality across the wound by migrating along a newly formed polarised vasculature, which develops prior to SC migration (*Cattin et al., 2015*; *Cattin and Lloyd, 2016*). However, the collective migration of SCs as cords is an adaptive process, as SCs cultured alone exhibit contact inhibition of locomotion (CIL) (*Parrinello et al., 2010*), a process in which two cells repulse each other upon contact, resulting in the separation of cells (*Abercrombie and Heaysman, 1953*; *Abercrombie and Heaysman, 1954*; *Carmona-Fontaine et al., 2008*; *Davis et al., 2015*). Following an injury, however, SC collective migration is triggered following interactions with fibroblasts that come into contact with SCs as they enter the wound site (*Parrinello et al., 2010*). This heterotypic interaction with fibroblasts transforms SC behaviour from repulsive to attractive, a process mediated by ephrinB/EphB2 signalling inducing a Sox2-mediated re-localisation of N-cadherin to the site of cell–cell junctions, which results in SCs migrating as cellular cords (*Parrinello et al., 2010*).

Recently, CIL has been shown to play a role in the dispersal of cells during development, with CIL promoting the spread of cells through tissues (*Davis et al., 2012*; *Villar-Cerviño et al., 2013*). Moreover, CIL is also important for providing an outward force during collective cell migration, in that CIL regulates the polarisation of cells by inhibiting local protrusions at the site of cell–cell contact and inducing protrusion formation at the free edge, thus promoting outward migration (*Abercrombie and Heaysman, 1953*; *Abercrombie and Heaysman, 1954*: *Carmona-Fontaine et al., 2008*; *Davis et al., 2015*; *Scarpa et al., 2015*; *Theveneau et al., 2010*). This implies that collective migration requires the maintenance of a CIL signal in the presence of a stronger adhesive signal, but how this can be achieved is poorly understood.

Here we show that N-cadherin mediates both the adhesive and repulsive forces required for collective SC migration but via two distinct mechanisms; with adhesion dependent on the Sox2-stabilised, N-cadherin adherens junction complex, while repulsion is the result of N-cadherin presenting a Slit2 and Slit3-repulsive signal. Moreover, because of this dual role, inhibiting the CIL signal results in the formation of tight clusters of non-migratory cells. Consistent with this, in ex vivo slices from mice that had undergone sciatic nerve injury, inhibition of the Slit2-repulsive signal impaired collective SC migration, resulting in cell clusters that appeared to lack directionality. The ability of N-cadherin to simultaneously regulate adhesion and CIL shows how these opposing processes can be coordinated to achieve outward collective migration and that CIL signals may present an attractive target for inhibiting unwanted collective cell migration.

## Results

### N-cadherin is required for contact inhibition of locomotion between Schwann cells

We previously showed that EphB2 activation of Sox2 results in the clustering of SCs, but video analysis indicated that these clustered cells maintain CIL as the cells appeared to be repulsing each other within the cluster (*Parrinello et al., 2010*). This is consistent with the migration of SC cords during nerve regeneration, which would be predicted to require a force, such as CIL, to drive migration forward (*Carmona-Fontaine et al., 2008*; *Haeger et al., 2015*; *Roycroft et al., 2018*; *Theveneau et al., 2010*). N-cadherin has been implicated in CIL in other cell types (*Scarpa et al., 2015*; *Tanaka et al., 2012*; *Theveneau et al., 2010*), so we addressed whether N-cadherin was also required for the regulation of SC collective migration. To do this, we initially blocked N-cadherin expression using two independent siRNAs (*Figure 1—figure supplement 1a and b*) and performed a scratch assay (*Figure 1a–e*). Time-lapse microscopy showed that while scrambled control siRNA-treated cells migrated in a directional manner to efficiently close the gap, N-cadherin-knockdown cells closed the

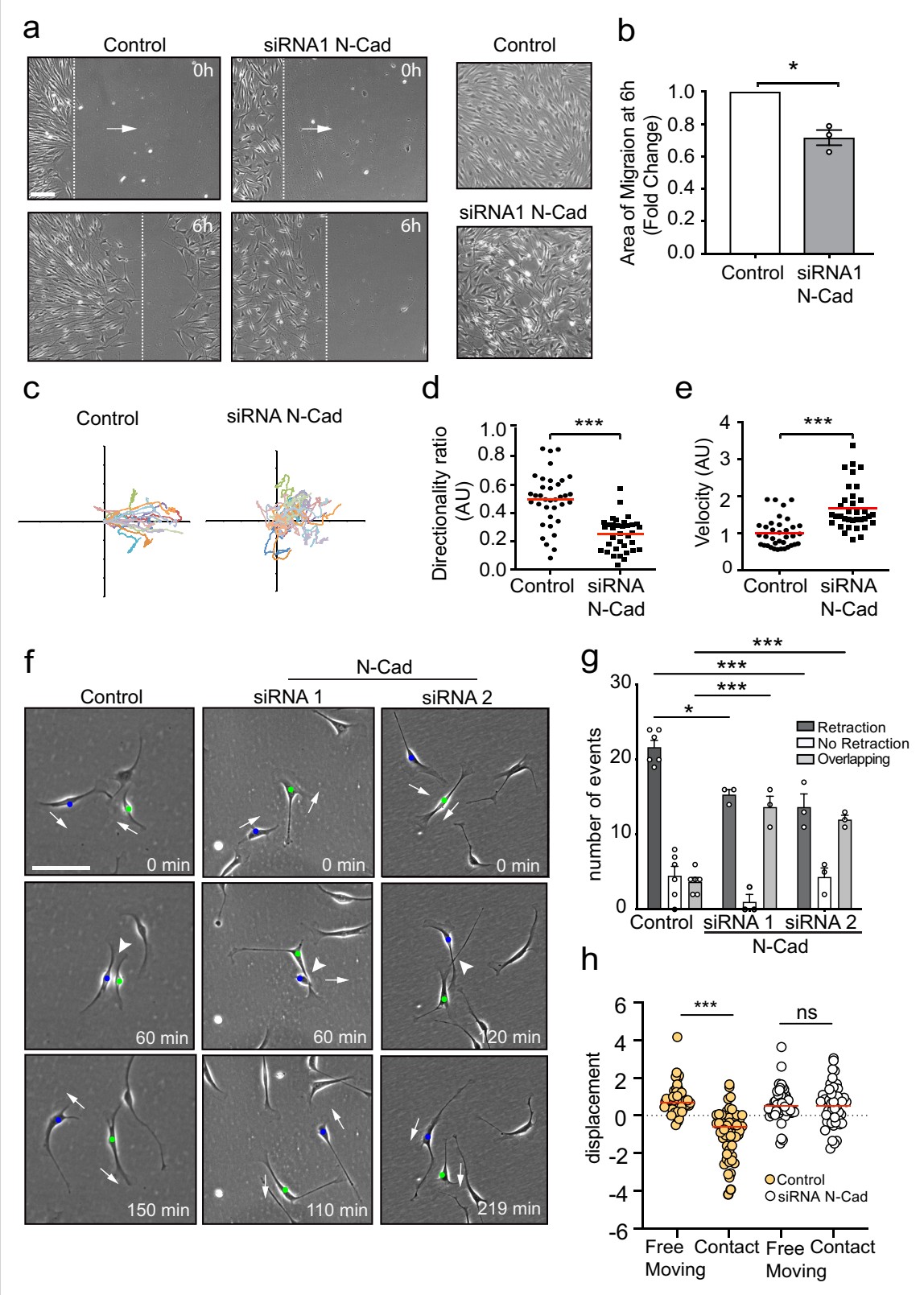

**Figure 1.** N-cadherin is required for contact inhibition of locomotion between Schwann cells (SCs). (**a**) Representative still images comparing the collective migration of siRNA Scrambled Control or N-cadherin (N-Cad) KD SCs in a wound healing assay. The dashed lines indicate the leading edge of migration at 0 hr and 6 hr. Arrows indicate the direction of migration. Zoomed images on the right show control SCs form a monolayer, whereas N-Cad KD SCs exhibit loss of cell:cell recognition with cells growing on top of each other. Scale bar = 100 µm. (**b**) Quantification of the area of migration

*Figure 1 continued on next page*

*Figure 1 continued*

of N-Cad knockdown SCs normalised to controls from (**a**). Data is presented as mean ± SEM and represent three independent experiments. p-values were calculated by using a two-tailed unpaired *t*-test with Welch's correction. (**c**) A representative graph of three independent experiments showing the trajectories of control or N-Cad knockdown cells treated with siRNA1. n = 36 and 35 for control and N-Cad knockdown cells, respectively. (**d, e**) Quantification of the directionality ratio (**d**) and the velocity (**e**) from the cells tracked in (**c**). The red line indicates the mean. p-values were calculated using unpaired two-tailed *t*-tests. (**f**) Representative time-lapse images of a contact inhibition of locomotion (CIL) assay, showing control or N-Cad knockdown cells, treated with siRNA1 or siRNA2 that repulsed or overlapped respectively (*Figure 1—video 1*). The blue and green dots indicate the two interacting cells. Arrows indicate the direction of migration. Scale bar = 100 μm. (**g**) Quantification of (**f**). Data are representative of n = 3 independent experiments and presented as mean ± SEM. p-values were calculated using a two-way ANOVA followed by Sidak's test for multiple comparisons. *p<0.05, **p<0.01, ***p<0.001. Note that random protrusions and retractions are produced by SCs in all directions that leads to a high number of apparent repulsion events, even when loss of CIL has occurred. (**h**) siRNA Scrambled cells (orange dots) exhibit a change in direction following contact (n=54) compared to free moving controls (n=58), indicative of CIL. N-cadherin knockdown (white dots) results in loss of CIL, with no difference in the displacement index between free moving (n=53) or contacting (n=53) cells. p-values were calculated using the Wilcoxon rank sum test. ***p<0.001.

The online version of this article includes the following video, source data, and figure supplement(s) for figure 1:

**Source data 1.** Excel spreadsheet containing data used to generate graphs in *Figure 1*.

**Figure supplement 1.** N-cadherin is required for contact inhibition of locomotion between Schwann cells (SCs).

**Figure supplement 1—source data 1.** Original file for the western blot analysis of N-cadherin KD in *Figure 1—figure supplement 1a* (N-cadherin).

**Figure supplement 1—source data 2.** Labelled file for the western blot analysis of N-cadherin KD in *Figure 1—figure supplement 1a* (N-cadherin).

**Figure supplement 1—source data 3.** Original file for the western blot analysis of loading control in *Figure 1—figure supplement 1a* (ERK).

**Figure supplement 1—source data 4.** Labelled file for the western blot analysis of loading control in *Figure 1—figure supplement 1a* (ERK).

**Figure 1—video 1.** N-cadherin mediates contact inhibition of locomotion (CIL).

https://elifesciences.org/articles/88872/figures#fig1video1

---

gap more slowly as they migrated in multiple directions over each other and with a lack of persistent migration towards the gap (*Figure 1a–d*). This difference was not due to a defect in migration speed as individual N-cadherin-knockdown cells migrated more rapidly (*Figure 1e*), suggesting that N-cadherin was required for the cell-contact-dependent process driving outward migration. Consistent with this, confluent control SCs form a monolayer and stable junctions, whereas N-cadherin knockdown cells were unable to form junctions and grew on top of each other (*Figure 1a*, *Figure 1—figure supplement 1b and c*). This indicated N-cadherin knockdown cells had lost the ability to recognise each other, and that the loss of contact-dependent outward migration observed in the collective migration assay may be due to loss of CIL.

To study the role of N-cadherin in homotypic CIL between SCs, we assessed CIL upon cell–cell contact of SCs cultured at low density. To quantify this, we looked at single-cell interactions and determined three different responses of the protrusions after the initial contact was made: *retraction*, SCs retract their protrusions and change direction of migration; *no retraction*, SCs interact for longer than 5 hr and do not change their direction or migrate away following contact; and *overlapping*, SCs continue to migrate following contact with their protrusions or cell bodies migrating on top of each other.

Live imaging of siRNA scrambled control SCs showed that SCs migrate in multiple directions, making frequent, seemingly random, protrusions and retractions in multiple directions. However, upon contact with another SC, the SC nearly always retracts the contacting protrusion, appears repulsed, and changes its direction of migration (*Figure 1—video 1*, *Figure 1f*, quantified in *Figure 1g*), behaviours that are features of CIL (*Abercrombie and Heaysman, 1953*; *Abercrombie and Heaysman, 1954*) and is consistent with what we observed previously (*Parrinello et al., 2010*). In contrast, while free-moving N-cadherin knockdown cells continued to make random protrusions and frequently changed direction, upon contact they behaved very differently to control cells, in that they continued to move forward upon contact and migrated on top of each other, a behaviour we termed overlapping. We did still observe retractions in the N-cadherin knockdown cells; however, we interpret this as background behaviour of cells that frequently change direction, even in the absence of contact. In contrast, the overlapping behaviour was rarely seen in control cells, showing that N-cadherin knockdown cells show no apparent recognition of each other and no CIL.

To confirm these results, we performed an additional quantification, which determines whether a cell changes its direction of movement following contact, compared to free-moving cells (*Paddock and Dunn, 1986*). In control cells, a significant difference between the displacement of free-moving and cells in contact was observed (*Figure 1h*) as control cells change direction after contact. In contrast, in N-cadherin knockdown cells, no difference was observed between free-moving and colliding cells, indicating that the cellular repulsion signal has been lost in these cells (*Figure 1h*). These results show that in addition to mediating adhesion between SCs, N-cadherin is also required for CIL between collectively migrating SCs.

## N-cadherin-dependent CIL is independent of the adherens junction complex

To understand how N-cadherin can mediate both cell:cell adhesion and CIL, we addressed whether both processes act via the N-cadherin adherens complex, which has previously been implicated in both processes (*Parrinello et al., 2010*; *Scarpa et al., 2015*; *Theveneau et al., 2010*). N-cadherin transmits adhesive forces between neighbouring cells by forming *trans*-homodimers which relay signalling via the well-characterised intracellular adherens complex to the actin cytoskeleton (*Brasch et al., 2012*; *Harris and Tepass, 2010*; *Peglion and Etienne-Manneville, 2013*; *Peglion et al., 2014*). We initially used time-lapse microscopy to analyse CIL between red-labelled, control-treated cells and green-labelled, N-cadherin knockdown cells in cocultures to determine the requirement for N-cadherin homodimers between the cells (*Figure 2a*, *Figure 2—video 1*). Surprisingly, while a N-cadherin knockdown cell (N2) invaded another N-cadherin knockdown cell (N1), the same cell was repulsed upon subsequent contact with a control cell (C2) (*Figure 2a*, quantified in *Figure 2b*). This suggested that N-cadherin is required to present a repulsion signal and induce repulsion but is not required for a cell to be repulsed. Consistent with this, when analysing the response of the siRNA scrambled control cells (C2) upon contact with an N-cadherin knockdown cell (N2), the majority of the control cells were not repulsed (*Figure 2a*, quantified in *Figure 2b*). However, the control cells did not invade the N-cadherin knockdown cells as the N-cadherin knockdown cells were repulsed and migrated away (*Figure 2—video 1*, *Figure 2b*). This shows that N-cadherin is required to present a repulsion signal and induce repulsion, but not to be repulsed (*Figure 2—video 1*, *Figure 2c*).

To further test this observation, we performed a dual-chamber assay, which assessed the ability of cells to form a boundary upon the closure of a gap. Consistent with the single-cell analysis, we found that whereas control cells formed a boundary upon gap closure and N-cadherin knockdown cells migrated on top of each other, N-cadherin knockdown cells were repulsed by the approaching control-treated cells and were unable to invade (*Figure 2d* and *Figure 2—figure supplement 1a*). These results confirmed that N-cadherin is only required to present the repulsive CIL signal, but not to be repulsed. Moreover, it suggests that N-cadherin mediates CIL independent of *trans*-homodimerisation, implying a distinct mechanism mediates CIL.

While the repulsion signal is independent of the *trans*-homodimerisation of N-cadherin, it remained a possibility that the proteins associated with the classical N-cadherin-adherens complex were required for the CIL signal. We therefore tested whether α-catenin and p120-catenin, which are known to physically connect N-cadherin to the actin cytoskeleton and regulate the stability of the N-cadherin adherens complex, are required for CIL (*Buckley et al., 2014*; *Peglion and Etienne-Manneville, 2013*; *Peglion et al., 2014*; *Pokutta and Weis, 2000*; *Rimm et al., 1995*; *Yao et al., 2014*). Time-lapse microscopy showed that efficient knockdown of α-catenin in SCs had no effect on CIL as the cells still repulsed each other despite an increase in cell velocity (*Figure 2e and f*, *Figure 2—figure supplement 1b and c*). However, consistent with previous reports, the connection of N-cadherin to the actin cytoskeleton appeared to be disrupted, as shown by the more cortical appearance of the actin cytoskeleton at N-cadherin cell–cell contacts (*Figure 2—figure supplement 1b*). Similarly, SCs depleted of p120 catenin still repulsed each other upon contact, although there was a slight but significant decrease in repulsive events in cells treated with siRNA2 (*Figure 2g and h*, *Figure 2—figure supplement 1d and e*). Consistent with the role of p120-catenin in regulating cadherin levels at the cell surface (*Davis et al., 2003*; *Miyashita and Ozawa, 2007*; *Nanes et al., 2012*; *Peglion and Etienne-Manneville, 2013*), we found a strong decrease in the levels of N-cadherin at the cell surface, although this was still concentrated at cell:cell contacts. However, this was greater in cells treated with the more efficient siRNA2, making it likely that a decrease in N-cadherin

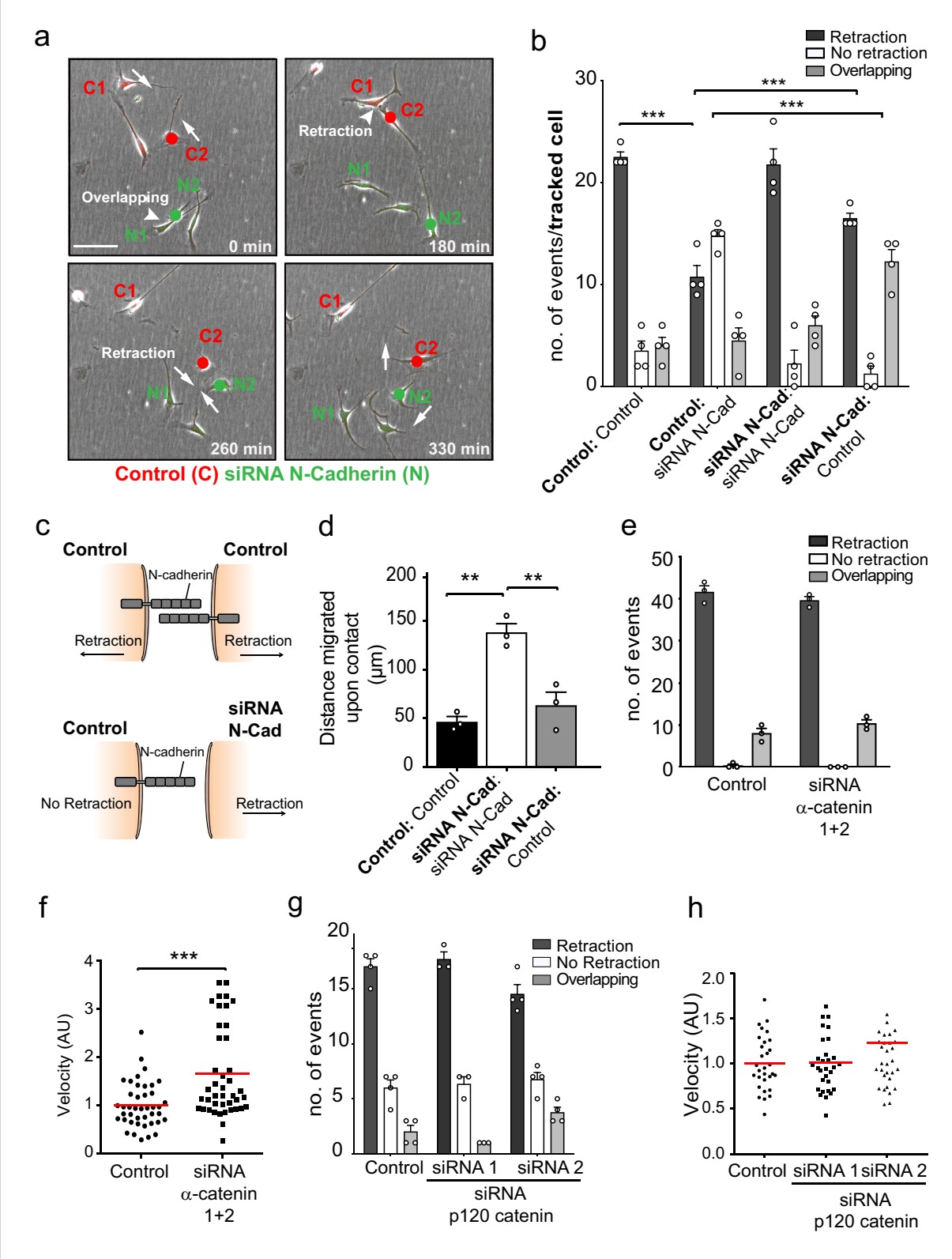

**Figure 2.** N-cadherin-dependent contact inhibition of locomotion (CIL) is independent of the adherens junction complex. (**a**) Representative time-lapse microscopy images of a CIL assay in which red fluorescence-labelled control cells (C1 and C2) were mixed with green fluorescent-labelled N-cadherin (N-Cad) knockdown cells (N1 and N2) (*Figure 2—video 1*). Cells of interest are indicated with a red or green dot for control and N-Cad knockdown cells, respectively. Arrows indicate the direction of migration. Scale bar = 100 μm. (**b**) Quantification of (**a**) n = 3 independent experiments. Graph

*Figure 2 continued on next page*

*Figure 2 continued*

represents the response of the cell being tracked (in bold) when it encounters the other cell type (not bold). Graph represents mean ± SEM; p-values were calculated using a two-way ANOVA followed by Sidak's test for multiple comparisons. (**c**) Schematic illustrating the mixing experiment in (**a, b**). When an siRNA scrambled control encounters another control cell, both cells express N-Cad and are repulsed away from each other. When a control cell makes contact with a N-Cad knockdown cell, only the control cell expressing N-Cad is capable of enacting a repulsive signal. (**d**) Quantification of the collective migration assay shown in *Figure 2—figure supplement 1a*, showing the distance migrated upon contact of control, N-Cad knockdown, or control and N-Cad knockdown cells following release from dual-chamber inserts. Data is presented as mean ± SEM and represents n = 3 independent experiments. p-values were calculated using a one-way ANOVA followed by Tukey's test for multiple comparisons. (**e**) Quantification of CIL in control and α-catenin knockdown cells. Data is presented as mean ± SEM from three independent experiments. p-values were calculated using a one-way ANOVA followed by Tukey's multiple comparisons test. (**f**) Quantification of the velocity of control and α-catenin knockdown cells. Red line indicates mean of n = 3 independent experiments. p-values were calculated by a two-tailed unpaired *t*-test. (**g**) Graph shows the quantification of CIL from n = 3 independent experiments. Data is presented as mean ± SEM. p-values were calculated using a one-way ANOVA followed by Tukey's multiple comparisons test. (**h**) Velocity of the tracks from control-treated cells (dots) and p120-catenin knockdown cells treated with siRNA1 (squares) or siRNA2 (triangles). Red lines indicate the mean of n = 3 independent experiments. p-values were calculated using a one-way ANOVA followed by Dunnett's post-test. *p<0.05, **p<0.01, ***p<0.001.

The online version of this article includes the following video, source data, and figure supplement(s) for figure 2:

**Source data 1.** Excel spreadsheet containing data used to generate graphs in *Figure 2*.

**Figure supplement 1.** N-cadherin-dependent contact inhibition of locomotion (CIL) is independent of the adherens junction complex.

**Figure supplement 1—source data 1.** Original file for the western blot analysis of α-catenin KD in *Figure 2—figure supplement 1c* (α-catenin).

**Figure supplement 1—source data 2.** Labelled file for the western blot analysis of α-catenin KD in *Figure 2—figure supplement 1c* (α-catenin).

**Figure supplement 1—source data 3.** Original file for the western blot analysis of loading control in *Figure 2—figure supplement 1c* (ERK).

**Figure supplement 1—source data 4.** Labelled file for the western blot analysis of loading control in *Figure 2—figure supplement 1c* (ERK).

**Figure supplement 1—source data 5.** Original file for the western blot analysis of p120 catenin KD in *Figure 2—figure supplement 1e* (p120 catenin).

**Figure supplement 1—source data 6.** Labelled file for the western blot analysis of p120 catenin KD in *Figure 2—figure supplement 1e* (p120 catenin).

**Figure supplement 1—source data 7.** Original file for the western blot analysis of N-cadherin in *Figure 2—figure supplement 1* (N-cadherin).

**Figure supplement 1—source data 8.** Labelled file for the western blot analysis of N-cadherin in *Figure 2—figure supplement 1e* (N-cadherin).

**Figure supplement 1—source data 9.** Original file for the western blot analysis of loading control in *Figure 2—figure supplement 1e* (ERK).

**Figure supplement 1—source data 10.** Labelled file for the western blot analysis of loading control in *Figure 2—figure supplement 1e* (ERK).

**Figure 2—video 1.** Contact inhibition of locomotion (CIL) is independent of *trans*-homodimerisation.
https://elifesciences.org/articles/88872/figures#fig2video1

at the cell surface was responsible for the small effect on CIL (*Figure 2g and h* and *Figure 2—figure supplement 1d and e*). Together these results indicated that N-cadherin-mediated CIL is both independent of *trans*-homodimerisation and acts independently of the adherens junction complex.

## The extracellular domain of N-cadherin is sufficient to mediate CIL

To investigate how N-cadherin mediates CIL, we tested which domains of N-cadherin were sufficient to rescue CIL in N-cadherin knocked-down cells. Western blotting showed that either full-length siRNA-resistant, tomato-tagged N-cadherin or mutants lacking the intracellular domain or the extracellular domain were expressed at similar levels (*Figure 3a and b*; *Shih and Yamada, 2012*). Confocal images showed that in SCs expressing the full-length exogenous N-cadherin construct, N-cadherin was localised at cell–cell junctions, and co-localised with α-catenin and p120-catenin (*Figure 3—figure supplement 1*). In contrast, in SCs expressing the intracellular domain of N-cadherin, N-cadherin did not form junctions and was observed at the membrane, co-localised with α-catenin and p120 catenin (*Figure 3—figure supplement 1*). Whereas in cells expressing the extracellular domain of N-cadherin, N-cadherin was present at the cell–cell junctions but was unable to recruit α-catenin and p120-catenin (*Figure 3—figure supplement 1*).

To analyse the effect of these constructs on CIL, we quantified the response of N-cadherin knockdown cells upon contact with N-cadherin construct-expressing cells at low density. Consistent with our earlier results, N-cadherin knockdown cells were repulsed by an N-cadherin expressing cell. As expected, while N-cadherin knockdown cells migrated over each other, they were repulsed upon contact with cells rescued with full-length N-cadherin (*Figure 3—video 1* and *Figure 3c*, quantified in *Figure 3d*), confirming that loss of CIL is specific to N-cadherin. In contrast, CIL was not rescued

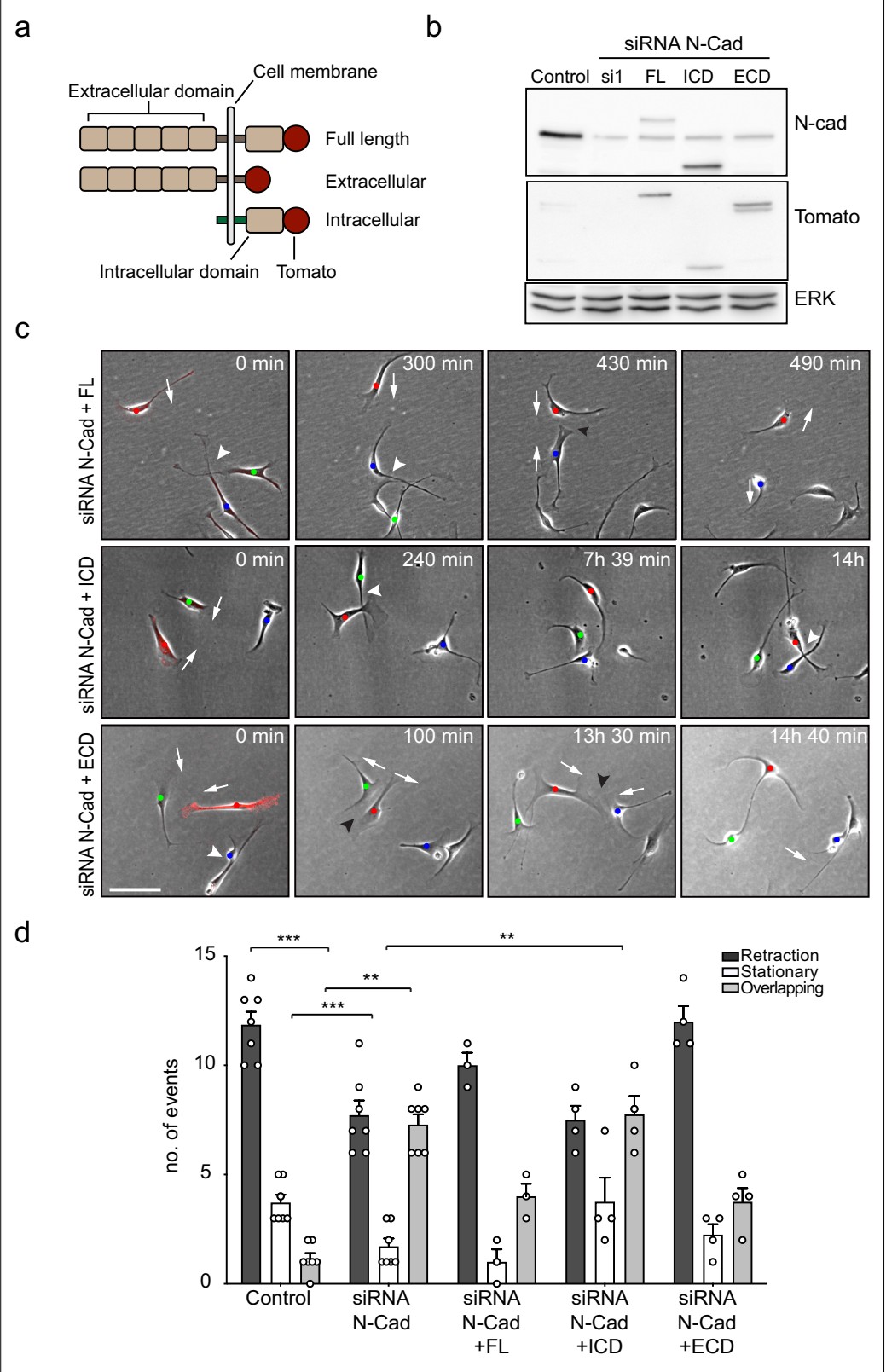

**Figure 3.** The extracellular domain of N-cadherin is sufficient to mediate contact inhibition of locomotion (CIL). (**a**) Schematic of N-cadherin (N-Cad) full-length, extracellular, and intracellular domains tagged with tomato at the C-terminus. The intracellular domain of N-Cad has an additional Lyn membrane-targeting sequence at the N-terminus to target it to the membrane. (**b**) Representative western blot using antibodies that recognise the

*Figure 3 continued on next page*

*Figure 3 continued*

C-terminus of N-Cad (127 kDa) and tomato (43 kDa), showing the expression levels of the constructs, 48 hr after knockdown of endogenous N-Cad using siRNA1. ERK (44, 42 kDa) was used a loading control. (**c**) Representative time-lapse microscopy images from a CIL assay of N-Cad knockdown cells transfected with siRNA-resistant full-length, intracellular domain (siRNA1+ICD) or the extracellular domain (siRNA1+ECD) of N-Cad tagged with tomato. Arrows indicate the direction of migration. Black arrowheads indicate repulsion events (siRNA1+full length and ECD). White arrowheads indicate overlapping events (siRNA1+Full length, ICD and ECD) (*Figure 3—video 1*). Cells of interest that are interacting are indicated by blue, red, and green dots. Scale bar = 100 μm. (**d**) Quantification of (**c**) full length (n = 3), the ECD and ICD of N-Cad (n = 4), and control and siRNA1 to N-Cad (n = 7). Graph shows the mean ± SEM. p-values were calculated using a two-way ANOVA followed by Sidak's test for multiple comparisons, **p<0.01, ***p<0.001.

The online version of this article includes the following video, source data, and figure supplement(s) for figure 3:

**Source data 1.** Original file for the western blot analysis of N-cadherin KD in *Figure 3b* (N-cadherin).

**Source data 2.** Labelled file for the western blot analysis of N-cadherin KD in *Figure 3b* (N-cadherin).

**Source data 3.** Original file for the western blot analysis showing the expression levels of the constructs in *Figure 3b* (tomato).

**Source data 4.** Labelled file for the western blot analysis showing the expression levels of the constructs in *Figure 3b* (tomato).

**Source data 5.** Original file for the western blot analysis of loading control in *Figure 3b* (ERK).

**Source data 6.** Labelled file for the western blot analysis loading control in *Figure 3b* (ERK).

**Source data 7.** Excel spreadsheet containing data used to generate graphs in *Figure 3*.

**Figure supplement 1.** The extracellular domain of N-cadherin is sufficient to mediate contact inhibition of locomotion (CIL).

**Figure 3—video 1.** The extracellular domain is sufficient to mediate contact inhibition of locomotion (CIL). https://elifesciences.org/articles/88872/figures#fig3video1

---

by SCs expressing the intracellular domain of N-cadherin (*Figure 3c and d*). Strikingly, CIL was fully rescued by SCs expressing the extracellular domain of N-cadherin showing that intracellular signalling by N-cadherin is not required to mediate CIL (*Figure 3c and d*). This result is consistent with our findings that the adherens junction complex is not required for N-cadherin mediated CIL, and together with our findings that N-cadherin is only required to present a repulsion signal, suggests that an additional co-repulsion signal may be required to mediate CIL in SCs.

## Glypican-4 and Slit2/Slit3 are required for CIL

We have previously shown that SCs are repulsed by fibroblasts in an ephrinB/EphB2-dependent manner (*Parrinello et al., 2010*), which induces cell clustering via activation of Sox2-dependent re-localisation of N-cadherin to cell–cell junctions. We thus reasoned that ephrinB/EphB2 was unlikely to be responsible for the homotypic CIL between SCs and this was confirmed in knockdown experiments (*Figure 4—figure supplement 1a*).

To identify the homotypic CIL signal, we performed a series of proteomic screens using N-cadherin or the extracellular domain of N-cadherin as bait, followed by co-immunoprecipitation and qualitative mass spectrometry analysis. In these analyses, peptides of Glypican-4, a glycosylphosphatidylinositol (GPI)-linked heparan sulphate proteoglycan, were detected in N-cadherin pull-down samples. Glypicans have previously been shown to play a role in axonal guidance and collective cell migration, so although not previously implicated in CIL, Glypican-4 was an interesting candidate (*Blanchette et al., 2015*; *Johnson et al., 2004*; *Venero Galanternik et al., 2015*). To test if Glypican-4 was required for mediating repulsive signals between SCs, we knocked down Glypican-4 (*Figure 4—figure supplement 1b*) and performed a CIL assay. Intriguingly, Glypican-4 knockdown cells did not repulse upon contact, nor did they invade (*Figure 4a*). Instead, upon contact it appeared that the cells adhered to each other and formed clusters (*Figure 4b and c*, *Figure 4—video 1*), with cell velocity prior to cluster formation unaffected (*Figure 4—figure supplement 1c*). This result was in stark contrast to N-cadherin knockdown cells, which migrated over the top of each other upon contact (*Figure 1f and g*). Moreover, the cell clusters appeared to behave differently to the clusters resulting from Sox-2 overexpression, with repulsive forces not evident between Glypican-4 knockdown cells upon contact,

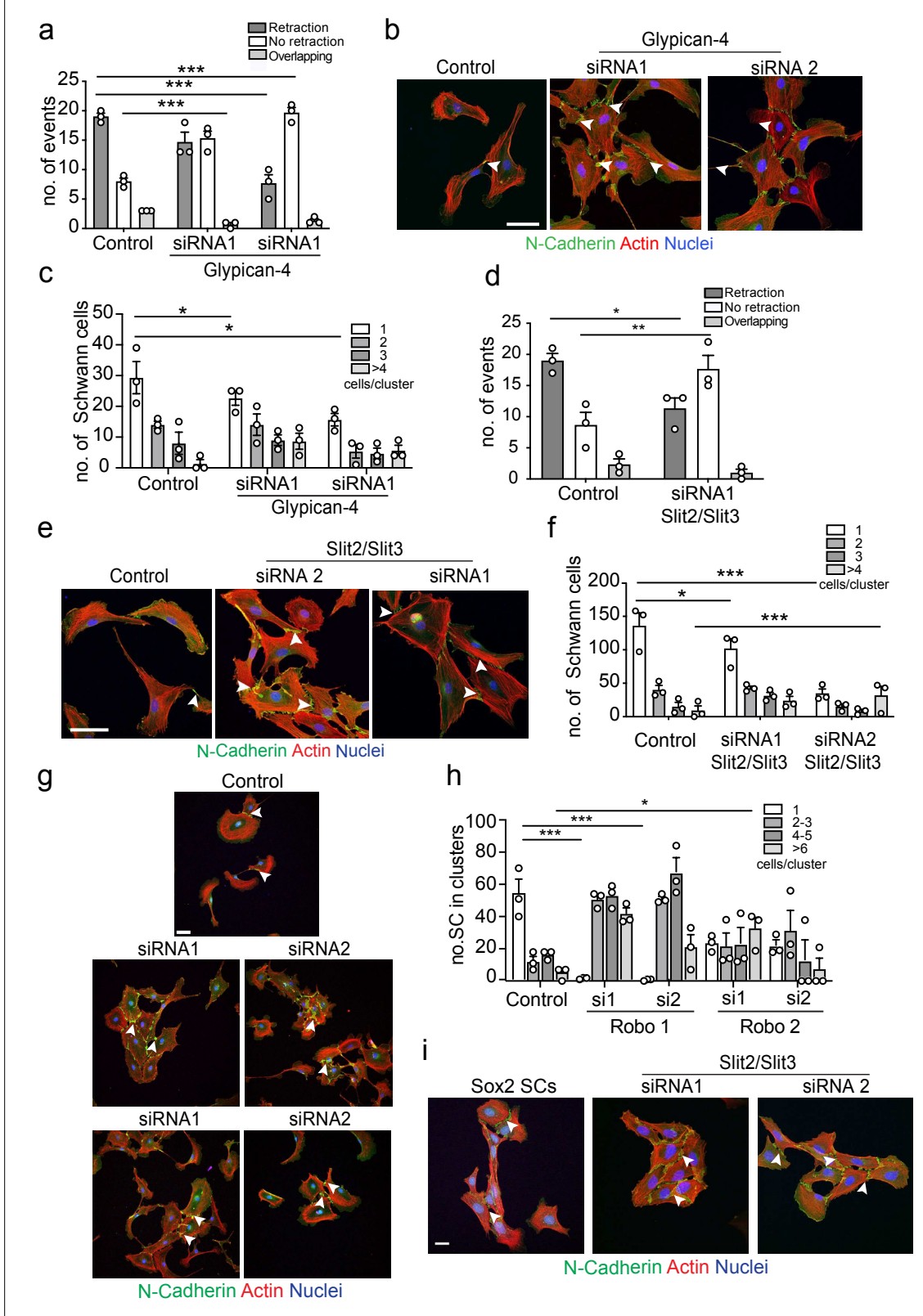

**Figure 4.** Glypican-4 and Slit2/Slit3 are required for contact inhibition of locomotion (CIL). (**a**) Quantification of a CIL assay, showing control or Glypican-4 knockdown cells, treated with siRNA1 or siRNA2, that are repulsed or not repulsed upon contact respectively (*Figure 4—video 1*). Data represents n = 3 independent experiments and mean ± SEM. p-values were calculated using a two-way ANOVA followed by Sidak's multiple comparisons test. (**b**) Representative confocal images showing control or Glypican-4 knockdown cells stained with phalloidin (red) and immunolabelled

*Figure 4 continued on next page*

*Figure 4 continued*

to detect N-cadherin (N-Cad) (green). Scale bar = 50 μM. Arrowheads indicate N-cadherin junctions. (**c**) Quantification of cluster formation in control or Glypican-4 knockdown Schwann cells (SCs) at 72 hr post-knockdown. Data represents n = 3 independent experiments and shows mean ± SEM. p-values were calculated using a two-way ANOVA followed by the appropriate post-test. (**d**) Quantification of CIL in control or Slit2/Slit3 knockdown cells treated with siRNA1 (n = 3, mean ± SEM) (*Figure 4—video 2*). p-values were calculated using a two-way ANOVA followed by Tukey's post-test for multiple comparisons. (**e**) Representative immunofluorescence images of control and Slit2/Slit3 knockdown SCs stained with phalloidin (red) and N-Cad (green). Scale bar = 50 μM. Arrowheads indicate N-cadherin junctions. (**f**) Quantification of SCs in clusters in control or Slit/Slit3 knockdown cells (n = 3, mean ± SEM). p-values were calculated using a two-way ANOVA followed by Tukey's test for multiple comparisons. (**g**) Representative confocal images showing control, Robo1, or Robo2 knockdown cells stained with phalloidin (red) and immunolabelled to detect N-cadherin (N-Cad) (green). Scale bar = 50 μm. Arrowheads indicate N-cadherin junctions (**h**) Quantification of the percentage of SCs in clusters following control or Robo1 or Robo2 knockdown (n = 3, mean ± SEM). p-values were calculated using a two-way ANOVA followed by Tukey's test for multiple comparisons. (**i**) Representative confocal images from n = 3 independent experiments of Sox2 overexpressing SCs compared to Slit2/Slit3 knockdown SCs stained with phalloidin to detect F-actin (red), antibodies to detect N-Cad (green), and Hoechst to detect nuclei (blue). Scale bar = 50 μm. *p<0.05, **p<0.01, ***p<0.001.

The online version of this article includes the following video, source data, and figure supplement(s) for figure 4:

**Source data 1.** Excel spreadsheet containing data used to generate graphs in *Figure 4*.

**Figure supplement 1.** Glypican-4 and Slit2/Slit3 are required for contact inhibition of locomotion (CIL).

**Figure supplement 1—source data 1.** Original gel for analysis of Robo1-3 expression in SCs in *Figure 4—figure supplement 1f*.

**Figure supplement 1—source data 2.** Labelled gel for analysis of Robo1-3 expression in SCs in *Figure 4—figure supplement 1f*.

**Figure supplement 1—source data 3.** Original gel for analysis of Robo 4 expression in SCs in *Figure 4—figure supplement 1*.

**Figure supplement 1—source data 4.** Labelled gel for analysis of Robo 4 expression in SCs in *Figure 4—figure supplement 1f*.

**Figure supplement 1—source data 5.** Original file for the western blot analysis of Slit2/3 KD in *Figure 4—figure supplement 1g* (Slit2).

**Figure supplement 1—source data 6.** Labelled file for the western blot analysis of Slit2/3 KD in *Figure 4—figure supplement 1g* (Slit2).

**Figure supplement 1—source data 7.** Original file for the western blot analysis of Slit2/3 KD in *Figure 4—figure supplement 1g* (Slit3).

**Figure supplement 1—source data 8.** Labelled file for the western blot analysis of Slit2/3 KD in *Figure 4—figure supplement 1g* (Slit3).

**Figure supplement 1—source data 9.** Original file for the western blot analysis of Slit2/3 KD in *Figure 4—figure supplement 1* (N-cadherin).

**Figure supplement 1—source data 10.** Labelled file for the western blot analysis of Slit2/3 KD in *Figure 4—figure supplement 1g* (N-cadherin).

**Figure supplement 1—source data 11.** Original file for the western blot analysis of loading control in *Figure 4—figure supplement 1g* (Vinculin).

**Figure supplement 1—source data 12.** Labelled file for the western blot analysis of loading control in *Figure 4—figure supplement 1g* (Vinculin).

**Figure supplement 1—source data 13.** Excel spreadsheet containing data used to generate graphs in *Figure 4—figure supplement 1*.

**Figure 4—video 1.** Glypican-4 is required for contact inhibition of locomotion (CIL) between Schwann cells.

https://elifesciences.org/articles/88872/figures#fig4video1

**Figure 4—video 2.** Slit2/3 mediates contact inhibition of locomotion (CIL) between Schwann cells (SCs).

https://elifesciences.org/articles/88872/figures#fig4video2

resulting in 'quieter' clusters that were no longer polarised towards outward migration (*Figure 4c*, *Figure 4—videos 1 and 2*). Consistent with this finding, N-cadherin was still present at the membrane in Glypican-4 knockdown cells with confocal images showing that N-cadherin accumulated at the cell–cell contacts, forming more pronounced junctions compared to the control (*Figure 4c*). This suggested that Glypican-4 is involved in the N-cadherin-dependent CIL signal and that in its absence SCs form more stable homotypic junctions, resulting in quieter clusters of cells. Consistent with this hypothesis, double knockdown of both Glypican-4 and N-cadherin resulted in cells that grew on top of each other, mimicking the N-cadherin knockdown phenotype (*Figure 4—figure supplement 1d and e*), highlighting the role for N-cadherin in presenting the repulsive signal, that is independent of its role at cell:cell junctions.

Glypicans are GPI-linked proteins that have been reported to act as co-receptors in several signalling pathways including Slit/Robo, Wnt, FGF, Hedgehog, and bone morphogenetic pathways or can modulate the accessibility of ligands. However, they are not thought to act as ligands themselves, indicating another signal may be required (*Sarrazin et al., 2011*; *Ypsilanti et al., 2010*). Of particular interest were the Slit/Robo signalling pathways, whose repulsive signals are known to play a role in axonal guidance (*Brose et al., 1999*; *Jia et al., 2005*; *Kaneko et al., 2010*; *Kidd et al., 1999*; *Nguyen-Ba-Charvet et al., 2004*; *Ypsilanti et al., 2010*). Previous studies have shown that of the three Slit genes, Slit2 and Slit3 are predominately expressed by SCs (*Carr et al., 2017*; *Chen et al., 2020*; *Wang et al., 2013*). We first confirmed this finding (*Figure 4—figure supplement 1f*)

and then performed knockdowns of both Slit2 and Slit3 in SCs (*Figure 4—figure supplement 1g*) and performed CIL assays. Initial studies showed a small loss of CIL with individual siRNAs, but the phenotype was not as strong as seen with Glypican-4, possibly because of compensation between the two molecules (data not shown). We therefore performed a double knockdown of Slit2 and Slit3 and found that Slit2/Slit3 knockdown cells behaved similarly to Glypican-4 cells in that they no longer repulsed upon contact, showing instead, increased levels of adhesion (*Figure 4d*, *Figure 4—video 2*), resulting in the formation of cell clusters with increased N-cadherin-mediated junctions compared to the control (*Figure 4e and f*).

To induce signalling, Slits can bind to Roundabout (Robo) receptors (*Ronca et al., 2001*; *Zhang et al., 2004*), which have been shown to mediate Slit-repulsive signals during neuronal development. Expression analysis showed that SCs express Robo 1,2, and 4 (*Figure 4—figure supplement 1f*), consistent with previous findings that Robo1 is expressed by SCs (*Carr et al., 2017*). To determine whether Slit acts via the Robo receptors, we performed knockdown experiments for Robo1 and Robo2 in SCs (*Figure 4—figure supplement 1h*) and assessed their ability to form clusters. Similarly, to the Slit2/Slit3 KDs, loss of either the Robo1 or Robo2 receptor in SCs resulted in a strong increase in cluster formation (*Figure 4g and h*), which resembled the quieter, round clusters observed with Slit2/Slit3 and Glypican-4 KDs. Together these results indicate that Slit2/Slit3 mediates the CIL signal through interactions with Robo1/2 and Glypican-4.

Following nerve injury, we have previously shown that SCs migrate collectively in polarised cords via activation of Sox2 and the resulting stabilisation of N-cadherin at the cell–cell junctions (*Cattin et al., 2015*; *Parrinello et al., 2010*). To compare the difference between these migratory SC clusters and the clusters induced by loss of Slit2/Slit3, we overexpressed Sox2 in SCs and compared them to Slit2/Slit3 KD SCs. Overexpression of Sox2 resulted in polarised clusters of SCs with increased N-cadherin at the cell–cell contacts (*Figure 4i*). In contrast, the clusters associated with Slit2/Slit3 knockdown had a very different morphology, consisting of more 'rounded' SCs, which lacked polarity, suggesting that the absence of Slit2/Slit3 signals between the cells results in cell clusters that lack an outward force to drive collective migration (*Figure 4i*, quantified in *Figure 4—figure supplement 1i*, and *Figure 4—video 2*).

## Loss of N-cadherin alters Slit2/Slit3 localisation at the cell surface

We have shown that both N-cadherin and Slit2/Slit3 are required for CIL between SCs. We hypothesised that this could be due to the loss of the repulsive Slit2/Slit3 signal in cells lacking N-cadherin. However, following N-cadherin knockdown, little change was detected in total Slit2 and Slit3 protein levels (*Figure 5—figure supplement 1a*) and mRNA levels of Slit2/3 and Glypican-4 were also not affected (*Figure 5—figure supplement 1b*), indicating a post-translational mechanism was involved. Co-immunoprecipitation experiments showed that Slit2 could be pulled down by N-cadherin (*Figure 5c*) and live imaging of SCs expressing tagged N-cadherin revealed that N-cadherin is a highly dynamic protein arriving in waves towards the cell's moving front, suggesting that perhaps a pool of N-cadherin was interacting with Slit2/Slit3 to facilitate its transport to the cell surface to mediate CIL (*Figure 5—figure supplement 1d* and *Figure 5—video 1*). To test this, we performed immunostaining of control and N-cadherin knockdown cells. Consistent with mRNA and protein analysis, total levels of Slit2 and Slit3 staining were unchanged following N-cadherin knockdown (*Figure 5—figure supplement 1e–g*). Confocal images of control SCs showed that Slit2 and Slit3 were localised at the membrane in both free-moving cells and cells in contact (*Figure 5—figure supplement 1*) and could be detected at lamellipodia (*Figure 5a–d*). Moreover, both ligands could be detected in numerous vesicles throughout the cytoplasm. In contrast, in N-cadherin knockdown cells, Slit2 and Slit3 were no longer observable at the cell surface but rather were sequestered in the perinuclear region (*Figure 5a and c* and quantified in *Figure 5b and d*). Importantly, Slit2 and Slit3 localisation could be restored by expression of the extracellular domain of N-cadherin (*Figure 5e and g* and quantified in *Figure 5f and h*). In contrast, Slit2 and Slit3 were still observed at the cell surface in Glypican-4 knockdown cells, suggesting that Glypican-4 does not appear to be involved in trafficking Slit2/Slit3 to the cell surface but more likely has a role in stabilising the cell surface presentation, as indicated by previous studies (*Ronca et al., 2001*; *Figure 5—figure supplement 1*). Together these results show that a pool of N-cadherin, via its extracellular domain, is required to facilitate Slit2 and Slit3 localisation at the cell surface, where, together with Glypican-4, they mediate CIL.

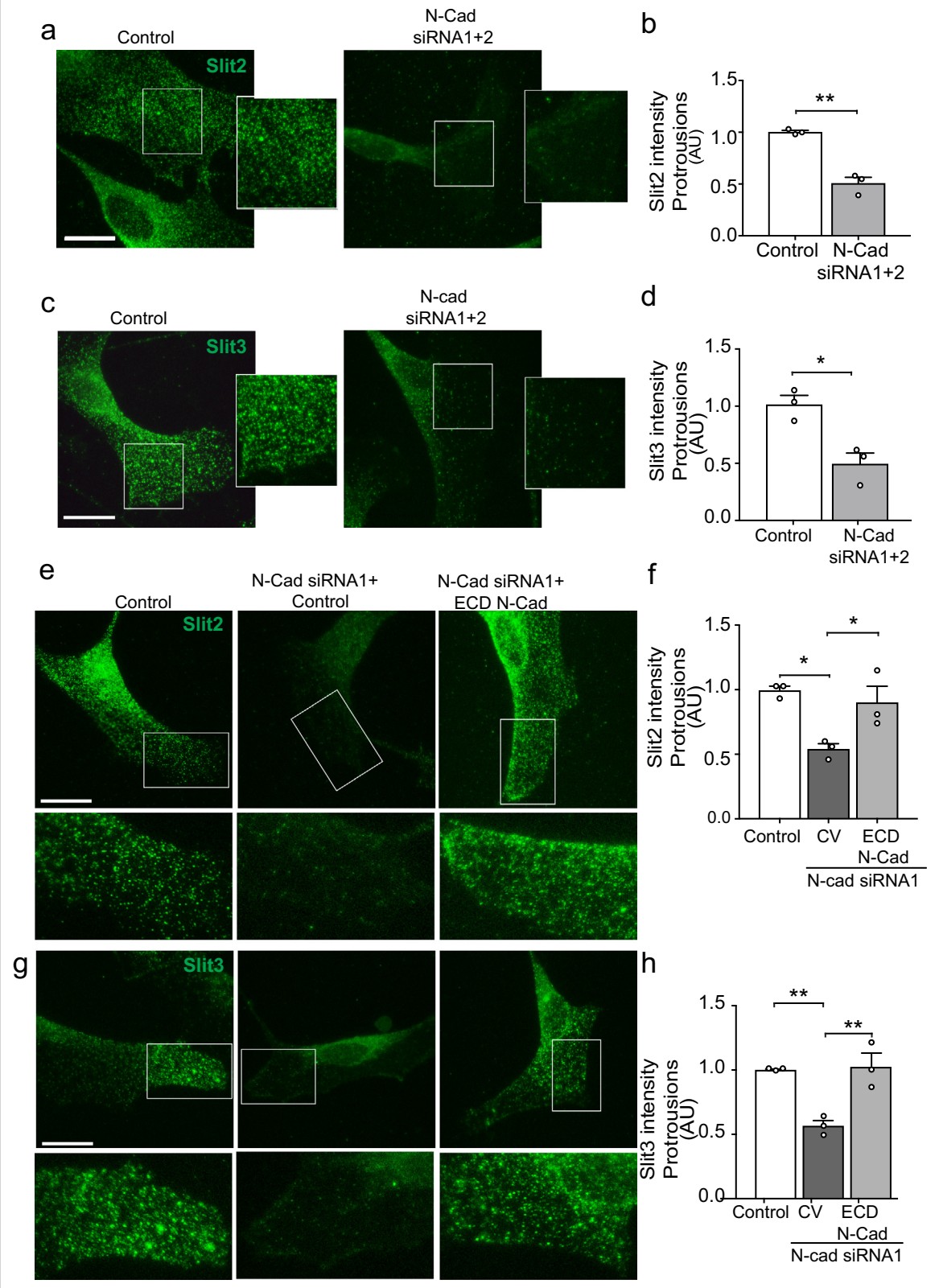

**Figure 5.** N-cadherin is required for the trafficking of Slit2/Slit3 to the cell surface. (**a–d**) Representative confocal images of control or N-cadherin (N-Cad) knockdown Schwann cells (SCs). Cells were labelled with antibodies to (**a**) Slit2 or (**c**) Slit3 (green) with quantification of Slit2 and Slit3 levels in the cell protrusions indicated by the boxes (n = 3, mean ± SEM for both conditions). Scale bars = 15 μm. p-values were calculated using an unpaired *t*-test. (**e**) Representative confocal images of rescue experiments in which SCs depleted of N-Cad were transfected with the N-Cad ECD tagged with tomato,

*Figure 5 continued on next page*

*Figure 5 continued*

or tomato control vector and immunolabelled to detect Slit2 (green). Scale bar = 15 µm. (**f**) Quantification of (**e**) (n = 3, mean ± SEM). p-values were calculated using a one-way ANOVA followed by Tukey's multiple comparisons tests. (**g**) As (**e**) but stained for Slit3 (green). (**h**) Quantification of (**g**) (n = 3, mean ± SEM). Scale bar = 15 µm. p-values were calculated using a one-way ANOVA followed by Tukey's multiple comparisons tests. *p<0.05, **p<0.01.

The online version of this article includes the following video, source data, and figure supplement(s) for figure 5:

**Source data 1.** Excel spreadsheet containing data used to generate graphs in *Figure 5*.

**Figure supplement 1.** N-cadherin is required for the localisation of Slit2/Slit3 at the cell surface.

**Figure supplement 1—source data 1.** Original file for the western blot analysis of N-cadherin KD in *Figure 5—figure supplement 1a* (N-cadherin).

**Figure supplement 1—source data 2.** Labelled file for the western blot analysis of N-cadherin KD in *Figure 5—figure supplement 1a* (N-cadherin).

**Figure supplement 1—source data 3.** Original file for the western blot analysis of N-cadherin KD in *Figure 5—figure supplement 1a* (Slit2).

**Figure supplement 1—source data 4.** Labelled file for the western blot analysis of N-cadherin KD in *Figure 5—figure supplement 1a* (Slit2).

**Figure supplement 1—source data 5.** Original file for the western blot analysis of N-cadherin KD in *Figure 5—figure supplement 1a* (Slit3).

**Figure supplement 1—source data 6.** Labelled file for the western blot analysis of N-cadherin KD in *Figure 5—figure supplement 1a* (Slit3).

**Figure supplement 1—source data 7.** Original file for the western blot analysis of N-cadherin KD in *Figure 5—figure supplement 1a* (Vinculin).

**Figure supplement 1—source data 8.** Labelled file for the western blot analysis of N-cadherin KD in *Figure 5—figure supplement 1a* (Vinculin).

**Figure supplement 1—source data 9.** Original file for the western blot showing the co-immunoprecipitation of either tomato or full-length N-Cad tagged with tomato, co-expressed with myc-tagged Slit2 in HEK cells in *Figure 5—figure supplement 1c* (Slit2).

**Figure supplement 1—source data 10.** Labelled file for the western blot showing the co-immunoprecipitation of either tomato or full-length N-Cad tagged with tomato, co-expressed with myc-tagged Slit2 in HEK cells in *Figure 5—figure supplement 1* (Slit2).

**Figure supplement 1—source data 11.** Original file for the western blot showing the co-immunoprecipitation of either tomato or full-length N-Cad tagged with tomato, co-expressed with myc-tagged Slit2 in HEK cells in *Figure 5—figure supplement 1c* (myc).

**Figure supplement 1—source data 12.** Labelled file for the western blot showing the co-immunoprecipitation of either tomato or full-length N-Cad tagged with tomato, co-expressed with myc-tagged Slit2 in HEK cells in *Figure 5—figure supplement 1c* (myc).

**Figure supplement 1—source data 13.** Original file for the western blot showing the co-immunoprecipitation of either tomato or full-length N-Cad tagged with tomato, co-expressed with myc-tagged Slit2 in HEK cells in *Figure 5—figure supplement 1c* (tomato).

**Figure supplement 1—source data 14.** Labelled file for the western blot showing the co-immunoprecipitation of either tomato or full-length N-Cad tagged with tomato, co-expressed with myc-tagged Slit2 in HEK cells in *Figure 5—figure supplement 1* (tomato).

**Figure supplement 1—source data 15.** Excel spreadsheet containing data used to generate graphs in *Figure 5—figure supplement 1*.

**Figure 5—video 1.** N-cadherin moves in waves towards cell contacts.

https://elifesciences.org/articles/88872/figures#fig5video1

## The Slit-repulsive signal is required for the efficient collective migration of Schwann cells

CIL can provide an outward force in collectively migrating cells (*Roycroft et al., 2018*; *Scarpa et al., 2015*; *Theveneau et al., 2010*). To test if the Slit CIL signal is important for the collective migration of SCs, we performed a collective migration assay with Slit2/Slit3 KD SCs and found this was sufficient to decrease the collective migration of these cells (*Figure 6a*, *Figure 6—figure supplement 1a*, and *Figure 6—video 1*). The ability to inhibit the CIL signal whilst maintaining cell:cell contacts has therapeutic potential in that rather than producing overlapping cells (N-cadherin inhibition) it should result in the formation of more tightly clustered, less migratory cells. Recombinant Slit2 (rSlit2) has been reported to induce a repulsive signal associated with a collapse of protrusions (*Wang et al., 2013*), and thus could potentially act in a dominant-negative manner by providing a uniform signal around cells. To test this, we initially analysed the effect of rSlit2 on CIL in low-density cultures and found that rSlit2 inhibited Slit-mediated repulsion signalling, as rSlit2-treated SCs no longer repulsed each other and instead formed clusters (*Figure 6b*, quantified in *Figure 6c*, and *Figure 6—video 2*), similar to that observed when Slit signalling was inhibited using siRNA (*Figure 4e*). Next, we performed a collective migration chamber assay in cells treated with rSlit2 and found that similar to the Slit2/Slit3 knockdowns, cells treated with rSlit2 migrated less efficiently than controls, with a slower closure of the gap (*Figure 6d*, *Figure 6—figure supplement 1b*, and *Figure 6—video 3*). By tracking the individual cells, we observed this behaviour was distinct from N-cadherin knockdown cells in that the cells migrated a shorter distance (*Figure 6e and f*) and appeared to adhere to each other, which prevented

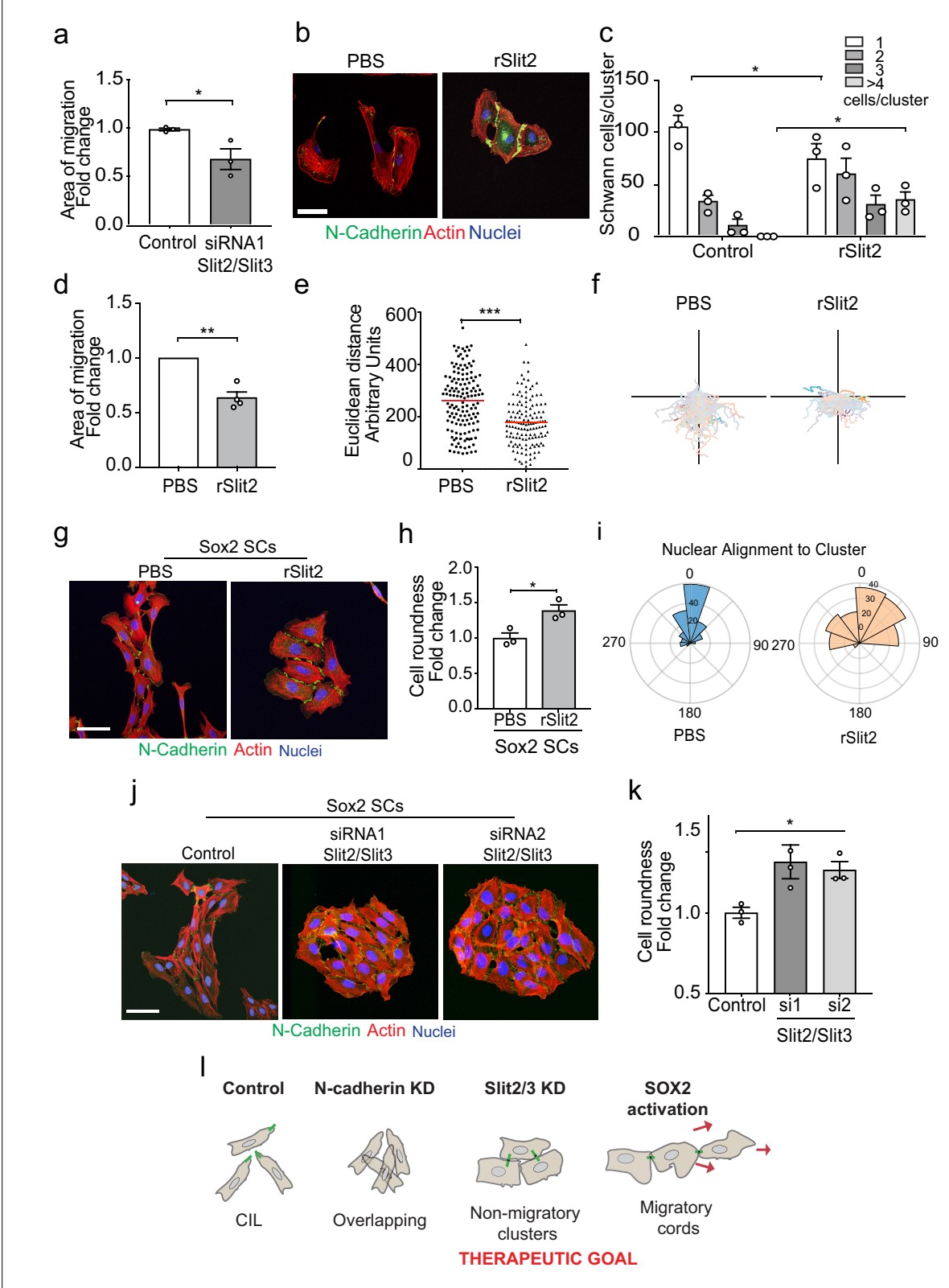

**Figure 6.** The Slit-repulsive signal is required for the efficient collective migration of Schwann cells (SCs) during nerve regeneration. (**a**) Quantification of the collective migration of control compared to Slit2/Slit3 knockdown SCs at 6 hr using a chamber assay (*Figure 6—video 1*). Data is normalised to control and presented as mean area ± SEM of n = 3 independent experiments. p-values were calculated using an unpaired *t*-test with Welch's correction compared to control. (**b**) Representative confocal images of SCs treated with recombinant-Slit2 (rSlit2) or PBS immunolabelled to detect

*Figure 6 continued on next page*

*Figure 6 continued*

N-Cadherin (N-Cad) (green) and co-stained with phalloidin (red) to detect F-actin and Hoechst to detect nuclei (blue). (n = 3). Scale bar = 50 μM. (**c**) Quantification of SC clusters from (**b**) (n = 3, mean ± SEM) (*Figure 6—video 2*). p-values were calculated using a two-way ANOVA followed by Sidak's test for multiple comparisons. (**d**) Graph shows the collective migration of SCs in a chamber assay treated with rSlit2 or PBS control (n = 3, mean area ± SEM). p-values were calculated using an unpaired *t*-test with Welch's correction compared to PBS controls. (**e**) Graph shows the Euclidean distance (shortest distance travelled) for cells in (**d**) at 24 hr (PBS n = 141, rSlit2 n = 138 from n = 3 independent experiments). The red line denotes the mean. p-values were calculated using an unpaired two-tailed *t*-test. (**f**) Graph shows tracks of individual cells in the collective migration assay quantified in (**d**) n = 3. (**g**) Representative confocal images from three independent experiments of Sox2-induced SC clusters treated with Shield for 24 hr and PBS or rSlit2 immunolabelled to detect N-Cad (green) and co-stained with phalloidin (red) to detect F-actin and Hoechst to detect nuclei (blue) (*Figure 6—video 3*). Scale bar = 50 μM. (**h**) Quantification of cell roundness of Sox2-induced SC clusters treated with PBS or rSlit2. (n = 3, mean ± SEM). Data was normalised to Sox2 controls and p-values calculated using an unpaired *t*-test. (**i**) Polar histograms showing alignment of nuclei within each cluster from PBS (n = 168) or rSlit2 (n = 166) treated Sox2 SCs as an indicator of polarisation. Angles closer to 0 represent more aligned nuclei in the PBS (blue) whereas they are more randomly distributed in the rSlit2-treated Sox2 SCs (orange). Data is representative of n = 3. (**j**) Representative confocal images of Sox2 SC clusters treated with control, or Slit2/Slit3 siRNAs and immunolabelled to detect N-Cad (green) and co-stained with phalloidin (red) to detect F-actin and Hoechst to detect nuclei (blue). Scale bar = 50 μm. (**k**) Quantification of the cell roundness of individual cells in Sox2-induced SC clusters treated with control or Slit2/Slit3 siRNA1 or 2 (n = 3, mean ± SEM). Data was normalised to Sox2 controls and p-values calculated using an unpaired two-tailed *t*-test. (**l**) Schematic illustrating that in control conditions SC exhibit CIL but upon KD of N-cadherin, CIL is lost and SCs become overlapping. Knockdown or inhibition of Slit2/Slit3 inhibits CIL but the persistence of N-cadherin expression results in the formation of non-migratory clusters. In contrast, SC cords in which Sox2 is activated maintain CIL signals which drive their collective migration. *p<0.05, **p<0.01, ***p<0.001.

The online version of this article includes the following video, source data, and figure supplement(s) for figure 6:

**Source data 1.** Excel spreadsheet containing data used to generate graphs in *Figure 6*.

**Figure supplement 1.** The Slit-repulsive signal is required for the efficient collective migration of Schwann cells (SCs).

**Figure supplement 1—source data 1.** Original gel for analysis of western blot showing pTuner empty vector SCs or pTuner Sox2 SCs response to Shield treatment at 24 hr in *Figure 6—figure supplement 1* (N-cadherin).

**Figure supplement 1—source data 2.** Labelled gel for analysis of western blot showing pTuner empty vector SCs or pTuner Sox2 SCs response to Shield treatment at 24 hr in *Figure 6—figure supplement 1c* (N-cadherin).

**Figure supplement 1—source data 3.** Original gel for analysis of western blot showing pTuner empty vector SCs or pTuner Sox2 SCs response to Shield treatment at 24 hr in *Figure 6—figure supplement 1c* (Sox2).

**Figure supplement 1—source data 4.** Labelled gel for analysis of western blot showing pTuner empty vector SCs or pTuner Sox2 SCs response to Shield treatment at 24 hr in *Figure 6—figure supplement 1c* (Sox2).

**Figure supplement 1—source data 5.** Original gel for analysis of western blot showing loading controls for pTuner empty vector SCs or pTuner Sox2 SCs response to Shield treatment at 24 hr in *Figure 6—figure supplement 1c* (Vinculin and α-tubulin).

**Figure supplement 1—source data 6.** Labelled gel for analysis of western blot showing loading controls for pTuner empty vector SCs or pTuner Sox2 SCs response to Shield treatment at 24 hr in *Figure 6—figure supplement 1* (Vinculin and α-tubulin).

**Figure supplement 1—source data 7.** Excel spreadsheet containing data used to generate graphs in *Figure 6—figure supplement 1*.

**Figure 6—video 1.** Slit2/3 are required for the efficient collective migration of Schwann cells (SCs).

https://elifesciences.org/articles/88872/figures#fig6video1

**Figure 6—video 2.** Recombinant Slit2 induces Schwann cell clustering.

https://elifesciences.org/articles/88872/figures#fig6video2

**Figure 6—video 3.** Slit2 is required for the efficient collective migration of Schwann cells (SCs).

https://elifesciences.org/articles/88872/figures#fig6video3

them from migrating forward, consistent with the loss of repulsion signal observed when Slit2/Slit3 signalling is inhibited at low density (*Figures 6b and c, and 4d and e*).

SCs migrate collectively as cords during nerve regeneration in response to a Sox2 signal that results in the formation of more stable N-cadherin junctions, which is dominant over the CIL signal (*Parrinello et al., 2010*). Loss of the CIL signal should therefore maintain these junctions while blocking the outward force required for collective migration. Consistent with this, we found that the addition of rSlit2 to Sox2-induced SC clusters resulted in clusters with a distinct morphology (*Figure 6g*), in that the clusters were rounder (*Figure 6—figure supplement 1d and e*), and that the cells within the cluster lacked polarisation away from the cluster, as measured by both cell roundness (*Figure 6h*) and nuclear alignment (*Figure 6i*, *Figure 6—figure supplement 1d*), suggesting a lack of repulsion between the cells (*Astin et al., 2010*). Moreover, loss of Slit2/Slit3 in Sox2-induced SC clusters resulted in a similarly profound change in cluster morphology with the formation of large, round, clusters that lacked polarity in contrast to the polarised clusters of cells that form in response to Sox2

(*Figure 6j*, quantified in *Figure 6k* and *Figure 6—figure supplement 1f and g*). Together these findings indicate that the outward force provided by Slit/Robo signalling is required for SC collective migration and that inhibiting this signal results in the formation of non-polarised clusters of cells with limited migratory ability (*Figure 6l*).

To confirm these findings within the context of the nerve environment, we developed a novel ex vivo explant system which allowed live visualisation of SC migration within the regenerating bridge region, following injury. This involved the use of transgenic mice in which eGFP is expressed in SCs (PLP-eGFP), allowing the visualisation of SCs within the complex environment of the nerve bridge (*Mallon et al., 2002*). Nerves were removed at day 5 following a transection injury, at which point SC cords are migrating into the new tissue formed at the injury site (the nerve bridge). The nerves were removed and embedded in buffered agarose. Sections (150 μm) of the bridge were then cut on a vibratome and placed into defined medium that allows the survival of the cells within the nerve bridge ex vivo for up to 24 hr (*Figure 7a*). Nerves were imaged prior to treatment (0 hr), before slices were incubated overnight with PBS or rSlit2. The following day (24 hr), slices were fixed and imaged to compare the effect of inhibition of Slit2 on migration. We observed that the structure of the injured nerve was maintained ex vivo, with the bridge comprising of densely packed nuclei, with regrowing axons migrating with SC cords along the newly formed vasculature (*Figure 7b*, *Figure 7—figure supplement 1a and b*), indicating that this is a powerful model in which we can visualise nerve regeneration ex vivo (*Cattin et al., 2015*; *Parrinello et al., 2010*).

At day 5 post-injury, at 0 hr, SC cords could be visualised beginning to migrate from the stumps into the newly formed nerve bridge (*Figure 7—figure supplement 1c*). Moreover, at 24 hr, the polarised cords of SCs had further migrated across the bridge in the PBS-treated explants, (*Figure 7c*, *Figure 7—figure supplement 1c*, and *Figure 7—video 1*). In contrast, treatment with rSlit2 resulted in an impairment of the collective migration of SCs, as evidenced by a decrease in the area of SC migration at 24 hr (*Figure 7c and d*). Moreover, within rSlit2-treated nerves, the SCs exhibited a distinct morphology in that there were significantly fewer SCs in polarised cords (*Figure 7f*) with the SCs instead, found in clusters, comparable in phenotype to those seen in vitro (*Figure 7g*). Further analysis of the SCs within the bridge, using Imaris to define the individual shape of the cells, showed that those treated with rSlit2 displayed a more rounded phenotype when compared to controls, consistent with the loss of CIL between the SCs, resulting in a loss of polarisation (*Figure 7h and i*). As an additional measurement of persistence of directionality, we measured the alignment of the following cells to the leader cell and found increased alignment in the control cells, consistent with polarised collective migration (*Figure 7j*, *Figure 7—figure supplement 1d*). Moreover, we also found the control cords were more aligned as they migrated across the wound site, whereas the rSlit2-treated explants appeared to lose this directionality, resulting in SCs migrating in more random directions into the bridge (*Figure 7k*). Together, these findings confirm the importance of the Slit2-mediated CIL signal for the effective migration of SC cords, which are essential for nerve regeneration. Moreover, these studies explain the apparently opposing dual roles for N-cadherin in the collective migration of SC during nerve regeneration, in that N-cadherin acts at SC cell:cell junctions to mediate the formation of migratory SC cords, whilst acting as a mediator of the repulsive Slit/Robo CIL signal to provide the force required to drive the outward migration of the SC cords that ensures successful nerve repair (*Figure 7l*).

## Discussion

SCs migrate collectively as cellular cords following the severing of a peripheral nerve, guiding regrowing axons back to their targets, and thereby promoting peripheral nerve regeneration (*Cattin et al., 2015*; *Cattin and Lloyd, 2016*; *Parrinello et al., 2010*; *Stierli et al., 2018*). Collective migration is a complex and continuously adapting process that requires the incorporation of multiple and opposing intercellular signals between individual cells within the group, as well as signals from the local environment such as chemotactic factors, which guide and induce migration, while repulsion signals from surrounding cells prevent the migrating cells invading tissues (*Haeger et al., 2015*; *Mayor and Etienne-Manneville, 2016*). During migration, intercellular-repulsive signals direct the migration of the cluster forward, whilst adhesive interactions are needed to maintain a cohesive group. How these seemingly opposing forces are coordinated to achieve collective migration remains mostly unclear (*Haeger et al., 2015*; *Mayor and Etienne-Manneville, 2016*).

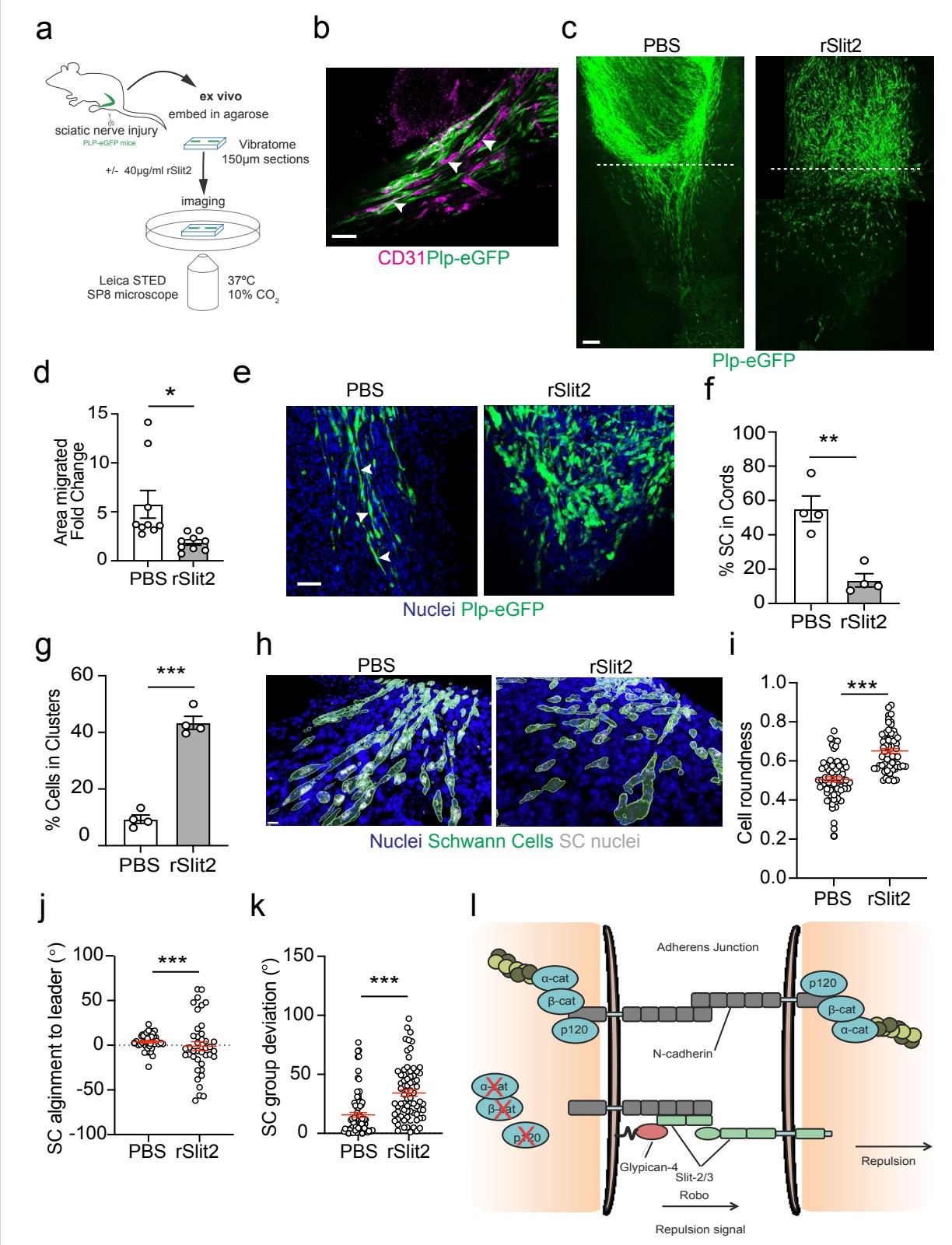

**Figure 7.** The Slit-repulsive signal is required for the efficient collective migration of Schwann cells (SCs) in the ex vivo nerve bridge. (**a**) Schematic showing ex vivo migration protocol. (**b**) Representative immunofluorescence images in untreated ex vivo explants showing plp-eGFP SCs (green) migrating along the vasculature (magenta) in the ex vivo bridge. Arrowheads indicate SCs in close contact with blood vessels. Scale bar = 50 μm. (**c**) Representative tile scan images of the nerve bridge explants following nerve transection in plp-eGFP (green) mice at 24 hr after treatment with PBS

*Figure 7 continued on next page*

Figure 7 continued

or rSlit (40 µg/ml). At 24 hr, PBS-treated explants showed the migration of aligned cords of SC into the bridge. In contrast, rSlit2-treated sections showed less SC migration, with groups of clustered, more round cells. Scale bar = 100 µm. (**d**) Graph showing quantification of SC migration at 24 hr following treatment with either PBS or rSlit2 in tile scan images. Data was expressed as the fold change in migration from 0 hr. Data are representative of n = 9 animals/group from n = 5 independent experiments. Data are represented as mean area ± SEM and p-values calculated by an unpaired *t*-test. (**e**) Representative ex vivo images of the nerve bridge 24 hr following treatment with PBS or rSlit2 to detect SC with endogenous plp-eGFP (green) and Hoechst (blue). Images show that PBS SC appear to migrate as directional cords whereas rSlit2-treated SC exhibit a rounder and more clustered phenotype (*Figure 7—video 1*). Scale bar = 50 µm. Images are representative of n = 4 animals/group from n = 3 independent experiments. (**f**) Quantification of (**d**) comparing the number of SC in cords following PBS or rSlit2 treatment and expressed as a percentage. p-values calculated by a Fisher's exact test (n = 4 animals/group mean ± SEM). (**g**) Quantification of (**d**) showing the number of SC in clusters following PBS or rSlit2 treatment and expressed as a percentage (n = 4 animals/group mean ± SEM). p-values calculated by a Fisher's exact test. (**h**) Representative 3D surface reconstructed images of nerve bridge explants following PBS or rSlit2 treatment labelled for SC (plp-eGFP, green), nuclei (blue), and SC nuclei (white). Images were used to quantify the sphericity of the migrating SC. Scale bar = 10 µm (**i**) Quantification of SC roundness from (**h**). PBS (n = 64) and rSlit2 (n = 65). p-values were calculated by an unpaired *t*-test with Welch's correction, n = 4 animals/group mean ± SEM (red). (**j**) Quantification of nuclear alignment of SCs to the leader cell within a cord or cluster. PBS (n = 41) and rSlit (n = 40). p-values were calculated by an unpaired *t*-test with Welch's correction, n = 3 animals/group mean ± SEM. (**k**) Quantification SC persistent migration. Graph represents how far SC groups have deviated from directed migration into the bridge. PBS (n = 72) and rSlit (n = 71). p-values were calculated by an unpaired *t*-test with Welch's correction, n = 4 animals/group mean ± SEM (red). (**l**) Cartoon representing the dual role of N-cadherin in SC collective migration. *p<0.05, **p<0.01, ***p<0.001.

The online version of this article includes the following video, source data, and figure supplement(s) for figure 7:

**Source data 1.** Excel spreadsheet containing data used to generate graphs in *Figure 7*.

**Figure supplement 1.** The Slit-repulsive signal is required for the efficient collective migration of Schwann cells (SCs) ex vivo.

**Figure 7—video 1.** Slit2 is required for the collective migration of Schwann cells (SCs) within a regenerating nerve.

https://elifesciences.org/articles/88872/figures#fig7video1

We previously showed that, similarly to other cell types such as astrocytes and neural crest cells (*Etienne-Manneville, 2014*; *Peglion et al., 2014*; *Theveneau et al., 2010*), the adherens junction molecule N-cadherin is required for SC collective migration by mediating the clustering of the SCs (*Parrinello et al., 2010*). In this study, we show that N-cadherin is also required for the repulsion signal, which provides an outward force for collective SC migration. N-cadherin has previously been shown to mediate the repulsion signal important for the collective migration of neural crest cells (*Becker et al., 2013*; *Scarpa et al., 2015*; *Theveneau et al., 2010*). However, despite neural crest cells and SCs belonging to the same lineage, we find that N-cadherin acts by a different mechanism to regulate CIL in SCs; CIL between neural crest cells was reported to require the *trans*-homodimerisation of N-cadherin that acted to inhibit local protrusion formation at cell–cell contacts via the adherens complex (*Becker et al., 2013*; *Scarpa et al., 2015*; *Theveneau et al., 2010*). In contrast, in SCs, while cell–cell adhesion is mediated by N-cadherin homodimers linked to the actin cytoskeleton via the adherens junction complex, N-cadherin mediates CIL by a distinct mechanism. This does not require *trans*-homodimerisation, is independent of the adherens junction complex, and only requires the extracellular domain. Instead, N-cadherin is required for the Slit2/Slit3 repulsion signal to be present at the cell surface (*Figure 7h*). The reasons for these differences remain unclear but could reflect that whereas SCs migrate in cordlike structures, neural crest cells migrate in much looser clusters, which appears to involve a co-attraction mechanism to maintain a migratory cluster (*Carmona-Fontaine et al., 2011*; *Shellard and Mayor, 2020*).

Slits are secreted axonal guidance molecules that are crucial for the development of the brain (*Blockus and Chédotal, 2014*; *Brose et al., 1999*; *Gonda et al., 2020*; *Jia et al., 2005*; *Kidd et al., 1999*; *Nguyen-Ba-Charvet et al., 2004*; *Ypsilanti et al., 2010*). However, while secreted, accumulating evidence suggests that the secreted forms can remain associated with the plasma membrane which would allow the local cell–cell signalling required for CIL (*Blockus and Chedotal, 2016*); indeed, it has been reported that CIL between fibroblasts involves Slit2/Robo4 signalling (*Fritz et al., 2015*). Here, we show that Slit2/Slit3 signalling is responsible for the homotypic CIL signal between SCs and that this signal is required for the efficient directional collective SC migration that is required for nerve repair. This is consistent with reports that Slit2, Slit3, and Robo1 are expressed by myelinating and non-myelinating SCs (*Carr et al., 2017*; *Wang et al., 2013*) and that a Slit2 signal can repulse a SC in culture (*Wang et al., 2013*). The ability of a secreted Slit signal to act as a repellent over both longer and short distances likely requires cell-specific mechanisms that regulate whether the Slit molecules

remain tethered to the cell surface (*Simpson et al., 2000a*; *Simpson et al., 2000b*). Our work shows that Slit-dependent CIL between SCs requires both the extracellular domain of N-cadherin and Glypican-4, and it therefore appears likely that this complex is important for the localisation and retention of Slit2/Slit3 at the cell surface. This is consistent with previous reports in which Glypican and other heparan sulphate proteoglycans appear to be important for local Slit signalling (*Liang et al., 1999*; *Piper et al., 2006*; *Ronca et al., 2001*). We show that Slit2/Slit3 protein localisation is dependent on N-cadherin, indicating a role for N-cadherin in the presentation of Slit2/3 at the cell membrane. Interestingly, we did not detect an obvious difference in Slit2 membrane staining between SCs in contact and free-moving SCs, which might suggest Slit2 is constantly presented at the cell surface, although this remains to be fully established (*Figure 5—figure supplement 1g*). The absence of Slit2/Slit3 in vesicles following N-cadherin knockdown would suggest N-cadherin is required for Slit2/Slit3 delivery to the membrane, rather than effects on recycling, but further work is required to establish the precise mechanism by which N-cadherin mediates the localisation of Slit2/Slit3 at the cell surface. Interestingly, Robo1/2 localisation does not appear to be dependent on N-cadherin expression as our mixing experiments (*Figure 2a and b*) show that N-cadherin knockdown cells are still repulsed by a control cell, implying that Robo1/2 are still capable of enacting a repulsion signal in the absence of N-cadherin.

Further complexities of the role of CIL in cell behaviour are indicated by the distinct mechanisms used to regulate heterotypic and homotypic CIL and how these signals can change the migratory behaviour of cells. SCs show complex migratory behaviours influenced by the microenvironment; SCs alone exhibit CIL, which we now show is a N-cadherin/Glypican4/Slit2/Slit3/Robo1/2-dependent process that repulse SCs from each other, which is perhaps important for the distribution of SCs along axons. Following an injury, however, SCs come into contact with fibroblasts at the injury site and the fibroblasts repulse SCs by an ephrinB/EphB2–dependent signal (*Parrinello et al., 2010*). Importantly, this signal also changes the behaviour of N-cadherin, resulting in the formation of more stable N-cadherin junctions via Sox2. Critical to this interaction is the dual role of N-cadherin in SC migration, with Sox2-mediated stabilisation of N-cadherin at the cell–cell contacts found to act dominantly over the CIL signal, resulting in the formation of cellular cords. However, within the cords, the Slit2/Slit3-dependent repulsion signal remains active, providing an outward force to drive the collective migration of these cells. While both N-cadherin and Slit2/Slit3 are important for the collective migration of cells, inhibition of these proteins has very distinct effects on the migration of SCs. So, whereas loss of N-cadherin results in the production of overlapping, highly migratory cells that lack collective behaviour, inhibition of Slit2/Slit3 results in the formation of tighter clusters of cells associated with enhanced N-cadherin junctions in which migration is inhibited (*Figure 6k*). This might suggest targeting of CIL signals would be a promising approach for inhibiting unwanted collective migration. Consistent with this, we showed that exogenous recombinant Slit2 (rSlit2) acts to inhibit SC CIL and inhibits the collective migration of these cells. Moreover, this effect could be reproduced in a powerful new ex vivo explant model, where we observed a similar phenotype with the failure to form cell cords associated with decreased migration, showing the Slit CIL signal is required for efficient SC migration within the regenerating nerve environment (*Figure 7b and c*). This failure to form polarised cords is consistent with reports that collective cell migration relies on the polarisation of the cell group in the direction of migration and the transmission of forces between the cell–cell contacts (*Capuana et al., 2020*; *Etienne-Manneville and Arkowitz, 2020*). The crucial role of collective migration has been well established in tumour cell migration and metastasis, with a role proposed for CIL in this process (*Abercrombie, 1979*; *Astin et al., 2010*; *Batson et al., 2014*; *Hwang et al., 2023*; *Jahedi et al., 2023*; *Paddock and Dunn, 1986*). Characterisation of CIL signals in distinct tumour types may therefore be a promising approach to identify new targets for inhibiting the invasion and spread of tumours.

## Materials and methods
### Animals
All animal work was performed in accordance with the UK Home Office legislation. Mice were group housed in temperature-controlled conditions on a 12 hr light–dark cycle with free access to food and water. To visualise SCs for ex vivo imaging, female and male (6–12 weeks old) plp-eGFP (*Mallon et al., 2002*) mice were used, in which SCs express eGFP.

## Sciatic nerve injury

Sciatic nerve injury was performed under aseptic conditions with isoflurane anaesthesia. Briefly, an incision was made, the right sciatic nerve exposed at the sciatic notch, and a full transection performed. The wound was then closed with surgical clips and the animal allowed to recover. For ex vivo imaging experiments, sciatic nerves were harvested at day 5 post injury.

## Ex vivo imaging

Sciatic nerves were harvested on day 5 post-sciatic nerve injury and collected on ice in EBSS dissection buffer (EBSS no phenol red [Gibco], 2 nM ascorbic acid, HEPES, 18 mM glucose, 1.3 mM $MgSO_4$, 2 nM $CaCl_2$). Next, nerves were placed in 3% low melting point agarose (Invitrogen) and the gel allowed to polymerise. Once the gel was set, 150 µm sections of nerve were cut on a Vibratome (Leica) in cold EBSS sectioning buffer (EBSS phenol red [Gibco], 2 mM ascorbic acid, 13.1 mM glucose, penicillin/streptomycin [Gibco]), sections were collected and stored on ice. Once sectioning was complete, nerves were mounted on 35 mm glass bottom dishes (MatTek) using a mix of 1.5% low melting point agarose and SC ex vivo media (Ham's F12 media [Gibco], 20 ng/ml ß-neuregulin [R&D Systems], 100 uM cAMP [Sigma], 5% horse serum, penicillin/streptomycin, 100 µg/ml transferrin, insulin [Sigma]). Tile scan images of the nerve bridge were acquired prior to treatment at 0 hr (Leica TCS SP8 STED) at 37°C, 10% $CO_2$. Sections were randomly divided into experimental groups and PBS or rSlit2 (40 µg/ml) was added blind to the ex vivo media and sections placed in the incubator at 37°C, 10% $CO_2$ for 24 hr. Explants were then fixed with 4% PFA and nuclei labelled with Hoechst before tile scan images were acquired to assess migration at 24 hr. For live imaging, sections were imaged on the Leica TCS SP8 STED at 37°C, 10% $CO_2$ every 15 min for up to 24 hr.

## Ex vivo image analysis

To quantify SC migration in ex vivo explants, maximum projection tile scan images from the bridge at day 5 were acquired using confocal microscopy (Leica TCS SP8 STED). To quantify the area of SC migration, a horizontal line was drawn to demarcate the boundary between the stump and the bridge and the area of SCs labelled with plp-eGFP was drawn using FIJI at 0 and 24 hr on. Data were expressed as area fold change in migration from 0 hr for each individual animal. To calculate the percentage of SC in cords or clusters, sections were stained with Hoechst for 30' after fixation. Total SC nuclei were counted based on overlap with plp-eGFP and SC morphology, and the number of SC in cords or clusters recorded. A cord was defined as two or more aligned SC migrating into the bridge. A cluster was defined two or more non-polarised SC. In order to calculate cell roundness, images of the bridge following 24 hr of treatment were opened on Imaris (V9.1.2). Surfaces were created for SCs (plp-eGFP) and nuclei (Hoechst), and segmented so that individual SCs could be identified in the nerve bridge, and the values for cell sphericity recorded. The same surface creation parameters were used for all images within an experiment, with minor adjustments in intensity threshold made to compensate for intensity differences between experiments. To calculate ex vivo SC persistence more directly, the same surface images were opened in FIJI and the angle of SC cords was measured relative to the stump and the angle of deviation from directional migration calculated. The same images were used to calculate the persistence of migration within each cord, with the angle of the nuclei measured in FIJI and calculated relative to the angle of the nuclei of the leading cell as an indicator of cell persistence (*Figure 7—figure supplement 1d*).

## Cell culture

Primary rat SCs were extracted from sciatic nerves of Sprague–Dawley rats at postnatal day 7 as previously described (*Mathon et al., 2001*). SCs were cultured on poly-L-lysine (PLL)-coated dishes in Dulbecco's Modified Eagle's Medium (DMEM, Lonza) supplemented with 3% fetal bovine serum (FBS, BioSera), 1 µM forskolin (Abcam), 200 mM L-glutamine (Gibco), GGF, 100 µg/ml kanamycin (Gibco), and 800 µg/ml gentamicin (Gibco) and maintained in 10% $CO_2$ at 37°C. HEK293T cells were cultured in DMEM (Lonza) supplemented with 10% FBS and 200 mM L-glutamine (Gibco).

Sox2-overexpressing SCs were produced using the Retro X ProteoTuner Shield system (Clontech). The Retro X ProteoTuner retroviral vector encodes a 12 kDa FKBP destabilisation domain (DD) that causes rapid degradation of the protein to which it is fused. The DD domain was fused to mouse Sox2 cDNA at the N-terminus. Accumulation of the DD-tagged protein was induced by addition of

Shield1 (Takara Bio) stabilising ligand to the media, which prevents the proteasomal degradation of the protein. For analysis of Sox2 clusters, Shield1 (200 nM) was added to Sox2-overexpressing SC or ProteoTuner SC controls and cells fixed 24 hr following treatment.

Cells for cell clustering or collective migration assays were pretreated with 2 µg/ml recombinant mouse Slit2 protein (rSlit2) (R&D Systems, 5444-SL-050) or PBS for 18 hr before time-lapse microscopy and/or fixation.

## Constructs

N-cadherin full length, the extracellular and intracellular domain of N-cadherin tagged with tomato on the C-terminus, were a kind gift from Prof. S. Yamada (*Shih and Yamada, 2012*). The myc-tagged Slit2 construct was a gift from Dr. V. Castellani (*Delloye-Bourgeois et al., 2015*).

## Antibodies

Primary antibodies were used that recognise N-cadherin (BD transduction); α-catenin (Sigma C2081); β-catenin (BD transductions 610920); p120-catenin (BD Transduction 61034); ERK1/2 (Sigma M5670), mCherry (Abcam ab183628; western blotting); mCherry (Life Technologies M11217; Immunoprecipitation), AKT 1/2/3 (Santa Cruz), Slit2 (Abcam ab134166; western blotting), Slit2 (Thermo Fisher Scientific PA531133; immunofluorescence), Slit3 (Sigma SAB2104337; immunofluorescence) Slit3 (R&D Systems AF3629; western blotting), Myc (Merck Millipore 05-724). Alexa Fluor secondary antibodies were obtained from Invitrogen. Horseradish peroxidase (HRP)-linked antibodies were obtained from GE Healthcare.

## Short interference RNA

All short interference RNA (siRNA) were purchased from QIAGEN, with All star control siRNA (QIAGEN) used a control. In brief, $10^5$ cells SCs were seeded on 6-well plates. The following day siRNA or control siRNA were mixed in plain DMEM with HiPerFect (QIAGEN) and incubated for 10 min at room temperature (RT) to allow complexes to form. Complexes were added to SCs for 16–18 hr, washed once with SC medium, and harvested or seeded for further experiments as appropriate.

## Mutagenesis

To disrupt the annealing of N-cadherin siRNA to the tomato-tagged N-cadherin full-length, or the extracellular domain of N-cadherin constructs, the NEB Q5 Site-Directed Mutagenesis Kit (NEB) was used to introduce silent mutations in the N-cadherin siRNA1 targeting sequence, which is located in the extracellular domain of N-cadherin, by mutating all four codons of the target sequence.

## Protein analysis

For protein extraction, cells were washed on ice with PBS, followed by snap-freezing at –80°C to break membranes and harvested in RIPA buffer (1% Triton X-100, 0.5% sodium deoxycholate, 50 mM Tris pH 7.5, 100 mM NaCl, 1 mM EGTA pH 8, 20 mM NaF, 100 µg/ml PMSF, 15 µg/ml aprotonin, 1 mM $Na_3VO_4$, 1/100 protease inhibitor cocktail). Cells were then lysed on ice for 30 min, vortexed every 10 min, and homogenised using a 26-gauge needle (Beckton Dickinson). Cell debris was pelleted by spinning at $750 \times g$ for 5 min at 4°C, and the supernatant was collected and quantified using the BCA assay (Pierce, Thermo Scientific).

## Western blotting

Western blotting was performed using Hoefer Scientific Instrument apparatus and Bio-Rad western blot electrophoresis system. 20–30 µg of protein was resolved using a sodium dodecyl sulphate-polyacrylamide gel electrophoresis (SDS-PAGE). Protein was transferred onto nitrocellulose membrane (Millipore-Immobilon) and blocked for 1 hr at RT using 5% milk-TBST. The membrane was incubated with primary antibodies overnight at 4°C. The following day, the membrane was washed three times with TBST, followed by incubation with the appropriate HRP conjugated secondary antibody. Subsequently, membranes were washed three times with TBS-T before detection of proteins of interest with Pierce-ECL western blot substrate (Thermo Scientific) or Luminata Crescendo Western HRP substrate (EMD-Millipore) on the Imagequant LAS 4000.

## Transfection of HEK293T cells

DNA was transfected using Attractene according to the manufacturer's instructions (QIAGEN). Briefly, HEK293T cells were seeded onto 60 mm plates 24 hr prior to transfection. 270 ng Myc-tagged Slit2 and 270 ng of tomato-tagged constructs of interest and 1.25 μg carrier vector DNA was incubated with Attractene in DMEM for 15 min at RT to allow complexes to form. Complexes were incubated for 2 hr at 37°C. Cells were harvested after 48 hr for co-immunoprecipitation (Co-IP).

## Co-immunoprecipitation

HEK cells were seeded at $1.2 \times 10^6$ onto 60 mm dishes and transfected as described above. Cells were scraped in NP40 buffer (50 mM Tris pH 7.5, 150 mM NaCl, 1% NP40 supplemented with 1/100 protease cocktail inhibitor [Sigma], 1/100 phosphatase inhibitor cocktail 2 and 3 [Sigma]), lysed on ice for 30 min. Debris was pelleted, by centrifuging at $750 \times g$ for 5 min at 4°C and the supernatant was collected into a fresh tube. Protein concentration was then quantified using BCA assay. All subsequent steps were performed at 4°C. Approximately 1 mg protein was pre-cleared using 10 μl of 50% protein G Beads (GE Healthcare) by rotating for 15 min. Beads were collected and discarded by centrifuging at $750 \times g$ for 1 min. Pre-cleared supernatant was incubated with primary antibodies to tomato (7 ug of anti-mCherry [Life Technologies]) or rat IgG as a control (Life Technologies) for 2 hr, rotating. The antibody-protein complexes were isolated by rotating the mixture with 50 μl of 50% Protein G beads for 1 hr. Beads were washed four times with 500 μl NP40 buffer and were collected by centrifugation at $750 \times g$ for 1 min and transferred to a clean tube. Finally, the beads were resuspended in 30 μl of Laemmli buffer and boiled for 10 min at 95°C.

To identify the homotypic CIL signal, we performed a series of proteomic screens using N-cadherin or the extracellular domain of N-cadherin as bait, followed by co-immunoprecipitation and mass spectrometry analysis. To do this, cells which displayed repulsion and overlapping behaviours were used to identify a signal that was only present in the repulsing cells. The co-IP was performed on one sample per condition using 1 mg protein as input, and the proteins eluted from the beads using Laemmli buffer. The proteins were then separated on a gel, extracted, digested with trypsin, and analysed using the LTQ Orbitrap Velos Pro Mass-spec systems (Performed by Proteomics Facility, University of Dundee). The obtained data was analysed using the Mascot search engine using the UniProt DB with rat as the taxonomic filter (performed by the Proteomics Facility, University of Dundee).

## RNA extraction and qPCR

RNA extraction was performed using Tri-Reagent according to the manufacture's protocol. RNA concentration was determined using Nanodrop and 500–1000 ng of RNA was used to synthesise complementary DNA using Superscript II kit (Invitrogen). For RT-qPCR, the MESA Blue qPCR MasterMix Plus kit was used (Eurogentec).

## Migration assays

### Contact inhibition of locomotion assays

For repulsion assays, $8 \times 10^3$ or $4 \times 10^3$ control or siRNA-treated SCs were seeded onto PLL and laminin coated 6-well plates or 12-well plates, respectively. Cells were allowed to adhere for a minimum of 6 hr and then time-lapse microscopy was performed. Live imaging was performed using a Zeiss Axiovert 200M microscope or the Nikon GFP3 at 37°C with 5–10% $CO_2$. Images were taken every 10 min for up to 72 hr.

To analyse interactions between control and N-cadherin knockdown cells, cells were treated with 10 μM Cell Tracker Red CMTPX Dye or Green CMFDA dye (Invitrogen) for 30 min at 37°C and washed once with SC medium. Following 1 hr incubation with Cell Tracker, $4 \times 10^3$ control cells were mixed with $4 \times 10^3$ N-cadherin knockdown cells and incubated for 16 hr before time-lapse microscopy.

Volocity or FIJI software was used to quantify repulsion. Single cells that were not dividing were tracked until contact with another cell and the initial response upon contact recorded. Three types of events were defined: *retraction*, cells retract protrusions and change direction of migration; *n-retraction*, cells interact for longer than 5 hr and do not change the direction of migration; *overlapping*, cell migrate on top of another cell with their protrusion and/or cell bodies.

For mixing experiments, the response of control-treated cells was quantified upon contact with N-cadherin knockdown cells and vice versa. Similarly, for rescue experiments only the response of

N-cadherin knockdown cells was quantified upon contact with a cell expressing either GFP, full-length N-cadherin, the extracellular domain of N-cadherin, or the intracellular domain of N-cadherin. To quantify the displacement of a cell after contact, vector analysis was used to analyse interactions between migrating cells. Videos were opened in FIJI and the displacement of a migrating cell 15 min prior to (Vector A) and following a collision (Vector B) calculated (*Paddock and Dunn, 1986*). The difference in displacement between the vectors (Vector B-A) was calculated to analyse the difference between how far the cell has progressed and how far it would have migrated had it not encountered another cell. Free-moving cells which did not collide with other cells were also analysed over the same duration. Cells exhibiting CIL have a negative value indicating that the cell has changed direction following collision.

### Rescue experiments

Rescue experiments were performed in a two-step protocol; first, siRNA transfection was performed using 1 nM of N-cadherin siRNA1 as described above and incubated overnight. The following morning, the complexes were removed, and 4 hr later full-length N-cadherin, the extracellular domain of N-cadherin, or the intracellular domain of N-cadherin were transfected using Attractene (QIAGEN). Complexes were incubated with control or N-cadherin knockdown cells for 2 hr at 37°C. The following day, cells were seeded for repulsion assays and immunofluorescence.

Time-lapse microscopy was performed approximately 6 hr after seeding for 24 hr.

### Collective migration assays

For collective migration assays, two approaches were taken. Either (i) $1 \times 10^5$ cells were seeded onto laminin-coated plates and the following day, siRNA transfection was performed as described above. A scratch was then induced with a sterile tip, 48 hr after knockdown. The cells were gently washed twice with medium to remove any debris followed by time-lapse microscopy of the leading edges for 24 hr. Or (ii) $1.5 \times 10^4$ control or knockdown cells were seeded into the separate compartments of dual-chamber inserts (Ibidi), 24 hr after siRNA transfection. The following morning, each compartment was treated with CellTracker-Green or Cell Tracker Red CMTPX Dye, for 30 min. Subsequently, the chamber was removed, and cell migration imaged using time-lapse microscopy for 24 hr on the Nikon GFP3 at 37°C, 10% $CO_2$. In order to quantify collective migration, the area of cells was drawn using FIJI at 0, 6, or 24 hr on stills from time-lapse microscopy. The area migrated was calculated using the formula:

$$\text{Area migrated} = \text{Area}\,0 + \text{Xh} - \text{Area}\,0\,\text{h}$$

Data were expressed as area fold migration change relative to PBS or control siRNA.

## Cell tracking

Cells were tracked by their nucleus using Volocity software or the manual cell tracking plugin in FIJI for 8–12 hr (dividing cells were excluded). Velocity and directionality were then measured from the tracks using macro plugin in Excel as described in *Gorelik and Gautreau, 2014*.

## Cell clustering assays

Cell clustering assay was performed as described in *Parrinello et al., 2010*. Briefly, $3.5 \times 10^3$ cells treated with Glypican-4, Robo1/2, or Slit2/3 siRNA were seeded onto coverslips 48 hr after knockdown. Clusters were fixed 24 hr after seeding, and immunofluorescence performed as described below. For rSlit2-treated cells, $5 \times 10^3$ cells were seeded onto a 12-well plate and clusters quantified after 36 hr of treatment. The following clusters were defined, single cells, 2, 3, or >4 cells.

## Immunofluorescence

Cells grown on glass coverslips were fixed in 4% paraformaldehyde (PFA) supplemented with 1 mM $CaCl_2$ and 0.5 mM $MgCl_2$ to prevent disruption of calcium-dependent complexes such as N-cadherin, for 10 min at RT. Cells were then permeabilised with 0.3% Triton PBS for 10 min and blocked with 3% bovine serum albumin-PBS (BSA) for 1 hr at RT and incubated with primary antibodies overnight at 4°C. The following day, coverslips were washed with PBS, incubated with the appropriate secondary

antibodies for 1 hr at RT, and washed with PBS before mounting onto microscope slides using fluoro-mount-g (Southern Biotech). For labelling of Slit2 and Slit3, cells were fixed in 2% PFA supplemented with 1 mM CaCl$_2$ and 0.5 mM MgCl$_2$ and blocked in 3% BSA. All remaining steps were performed as described above. All images were acquired using an inverted Leica TCS SPE confocal microscope, with image processing and analysis performed using FIJI software.

## Quantification of immunofluorescence

To quantify Slit3 and Slit2 immunolabelling, the outline of the cells was drawn using the free-hand drawing tool in FIJI and the fluorescence intensity was measured using FIJI. The nucleus and perinuclear area were excluded from the quantification in order to calculate the intensity in the protrusions. In order to quantify the roundness of individual SCs in clusters or the roundness of the entire cluster, the following equation from *Rotty et al., 2017* was used:

$$Shape = 4\Pi \times Area / Perimeter^2$$

where the area and perimeter of each cell were measured by tracing their outline using FIJI from confocal images. Groups of SCs were considered to be in clusters if they consisted of >4 cells. To further quantify polarity of the clusters, images were opened in FIJI and the angle of the cluster in the direction of movement measured. The nuclei of each cell in the cluster was also measured, and the angle relative to the cluster axis calculated. Data was then imported into MATLAB to generate polar histogram plots.

## Statistics

Statistics were performed using GraphPad (Prism) software. Data is presented as mean ± SEM and is representative of at least n = 3 independent experiments. Data were analysed using a one-way or two-way ANOVA, followed by multiple comparisons tests, two-tailed unpaired Student's *t*-test with Welch's correction or Mann–Whitney test as appropriate and are detailed in the figure legends. In all cases, *$p<0.05$, **$<0.01$, ***$<0.001$.

## Acknowledgements

This work was supported by a programme grant from Cancer Research UK (C378/A4308), an MRC PhD studentship to JJAH and core support by MRC funding to the MRC LMCB University Unit at UCL, award code MC_U12266B. We would like to thank UCL Biological Services for helping with the maintenance of our animals, Giulia Casal for useful advice on image analysis and Imaris, Lucie Van Emmenis and Stella Kouloulia for assistance with surgeries, Liza Malong for her input regarding the ex vivo protocol, and the rest of the Lloyd lab for useful discussions.

## Additional information

### Funding

| Funder | Grant reference number | Author |
| --- | --- | --- |
| Cancer Research UK | C378/A4308 | Alison C Lloyd |
| Medical Research Council | Studentship | Julian JA Hoving |

The funders had no role in study design, data collection and interpretation, or the decision to submit the work for publication.

### Author contributions

Julian JA Hoving, Conceptualization, Formal analysis, Investigation, Methodology, Writing – original draft, Data curation, Visualization; Elizabeth Harford-Wright, Formal analysis, Investigation, Visualization, Methodology, Writing – original draft, Conceptualization, Data curation, Writing – review and editing; Patrick Wingfield-Digby, Mariana Campana, Toby Morgan, Victor Quereda, Investigation; Anne-Laure Cattin, Investigation, Methodology, Writing – review and editing; Alex Power, Investigation,

Writing – review and editing; Erica Torchiaro, Investigation, Methodology; Alison C Lloyd, Conceptualization, Resources, Formal analysis, Supervision, Visualization, Methodology, Writing – original draft, Project administration, Writing – review and editing, Funding acquisition

### Author ORCIDs
Elizabeth Harford-Wright http://orcid.org/0000-0001-7231-2108
Alison C Lloyd http://orcid.org/0000-0001-7712-1773

### Ethics
This study was performed in accordance with UK Home office legislation and in close consultation with animal care staff at the University College London (UCL), Biological Services Central Unit. All animal work was carried out under the UCL establishment licence (X7069FDD2) and all procedures performed were approved by the UK Home office in project licence (PP9833892).

### Decision letter and Author response
Decision letter https://doi.org/10.7554/eLife.88872.sa1
Author response https://doi.org/10.7554/eLife.88872.sa2

---

## Additional files

### Supplementary files
• MDAR checklist

### Data availability
All data are included in the manuscript, figures and figure supplements, and source data files.

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

# Appendix 1

## Appendix 1—key resources table

| Reagent type (species) or resource | Designation | Source or reference | Identifiers | Additional information |
|---|---|---|---|---|
| Other | Plp-eGFP | *Mallon et al., 2002* | | Available from Jackson Laboratories https://www.jax.org/strain/033357 |
| Cell line (*Rattus norvegius*) | Schwann cells | *Mathon et al., 2001* | SCs | |
| *Cell line (R. norvegius)* | Sox2 Schwann cells | This paper | | Created using the Proteotuner Shield System by Clontech |
| Cell line (*R. norvegius*) | ProteoTuner Schwann Cells | This paper | | Created using the Proteotuner Shield System by Clontech |
| Sequence-based reagent | αE-catenin siRNA1 | This paper | siRNA | AAGAACGCCTGGAAAGCATAA |
| Sequence-based reagent | αE-catenin siRNA2 | This paper | siRNA | CAACCGGGACTTGATATACAA |
| Sequence-based reagent | Cadherin-2 siRNA1 | This paper | siRNA | TCCCAACATGTTTACAATCAA |
| Sequence-based reagent | Cadherin-2 siRNA2 | This paper | siRNA | CAGTATACGTTAATAATTCAA |
| Sequence-based reagent | p120-catenin siRNA1 | This paper | siRNA | AGGTCAGATCGTGGAAACCTA |
| Sequence-based reagent | p120-catenin siRNA2 | This paper | siRNA | ATGCTCGGAACAACAAAGAGTTAA |
| Sequence-based reagent | Glypican-4 siRNA1 | This paper | siRNA | CCGACTGGTTACTGATGTCAA |
| Sequence-based reagent | Glypican-4 siRNA2 | This paper | siRNA | CGGTGTAGTTACAGAACTGTA |
| Sequence-based reagent | Slit2 siRNA1 | This paper | siRNA | ATCAATATTGATGATTGCGAA |
| Sequence-based reagent | Slit2 siRNA1 | This paper | siRNA | GACGACTAGACCGTAGTAATA |
| Sequence-based reagent | Slit3 siRNA1 | This paper | siRNA | AACGGCGGTGCCCAAAGAATT |
| Sequence-based reagent | Slit3 siRNA1 | This paper | siRNA | ATCGTGGAAATACGCCTAGAA |
| Sequence-based reagent | Robo1 siRNA1 | This paper | siRNA | AAGGGCGGCGAAAGAGTGGAA |
| Sequence-based reagent | Robo1 siRNA2 | This paper | siRNA | CCCGACTATAGAATGGTACAA |
| Sequence-based reagent | Robo2 siRNA1 | This paper | siRNA | CTCATTGGATTGTCCGGCTAA |
| Sequence-based reagent | Robo2 siRNA2 | This paper | siRNA | CTCGGACACTATCCTGCGGAA |
| Sequence-based reagent | N-cadherin siRNA1 targeting sequence | This paper | Forward primer | 5'-CACGATAAACAATGAGACTGGGGACATC-3' |
| Sequence-based reagent | N-cadherin siRNA1 targeting sequence | This paper | Reverse primer | Reverse primer 5'-AACATATTGGGTGAAGGTGTGCTGGG-3' |
| Sequence-based reagent | Slit1 forward | This paper | PCR primers | GCACTTGTCACAATGACCCT |
| Sequence-based reagent | Slit1 reverse | This paper | PCR primers | CCCTTCAAAGCCGGAAGGA |
| Sequence-based reagent | Slit2 forward | This paper | PCR primers | GTGTTAGAAGCCACGGGAAT |
| Sequence-based reagent | Slit2 reverse | This paper | PCR primers | GCGTCTGGTGTGAATGAGAT |
| Sequence-based reagent | Slit3 forward | This paper | PCR primers | GGATTATCGCAACAGATTCAG |
| Sequence-based reagent | Slit3 reverse | This paper | PCR primers | GGTCAGTGGTATATTCAGGG |
| Sequence-based reagent | Robo1 forward | This paper | PCR primers | AGGGGAGTCAGAATCTGCTT |
| Sequence-based reagent | Robo1 reverse | This paper | PCR primers | CCTCTGGACGTTCGTAACAG |
| Sequence-based reagent | Robo2 forward | This paper | PCR primers | TTGGATCAGAGGAGTCCCTG |
| Sequence-based reagent | Robo2 reverse | This paper | PCR primers | ACCCTTTAGAGGAGGCTGTT |

*Appendix 1 Continued on next page*

*Appendix 1 Continued*

| Reagent type (species) or resource | Designation | Source or reference | Identifiers | Additional information |
|---|---|---|---|---|
| Antibody | N-Cadherin (mouse monoclonal) | BD Transduction | 610920 | 1:1000 immunofluorescence western blot |
| Antibody | α-catenin (rabbit polyclonal) | Sigma | C2081 | 1:1000 immunofluorescence western blot |
| Antibody | β-catenin (mouse monoclonal) | BD Transduction | 610920 | 1:2000 immunofluorescence western blot |
| Antibody | p120-catenin (mouse monoclonal) | BD Transduction | 61034 | 1:2000 immunofluorescence western blot |
| Antibody | ERK1/2 (rabbit polyclonal) | Sigma | M5670 | 1:1000 western blot |
| Antibody | mCherry (rabbit polyclonal) | Abcam | ab183628 | 1:1000 western blot |
| Antibody | mCherry (rat monoclonal) | Life Technologies | M11217 | 1:1000 Immunoprecipitation |
| Antibody | Slit2 (rabbit monoclonal) | Abcam | ab134166 | 1:1000 western blot |
| Antibody | Slit2 (rabbit polyclonal) | Thermo Fisher Scientific | PA531133 | 1:1000 Immunofluorescence |
| Antibody | Slit3 (rabbit polyclonal) | Sigma | SAB2104337 | 1:1000 Immunofluorescence |
| Antibody | Slit3 (goat polyclonal) | R&D Systems | AF3629 | 1:1000 western blot |
| Antibody | Myc (mouse monoclonal) | Merck Millipore | 05-724 | 1:1000 western blot |
| Antibody | Alexa Fluor 546 Phalloidin | Life Technologies | A22283 | 1:1000 Immunofluorescence |
| Antibody | Goal anti-mouse Alexa Flour 488 | Thermo Fisher Scientific | A1100 | 1:1000 Immunofluorescence |
| Antibody | Rabbit IgG HRP | GE Healthcare | NA934V | 1:1000 western blot |
| Antibody | Mouse IgG HRP | GE Healthcare | NA931V | 1:1000 western blot |
| Antibody | Goat IgG HRP | R&D Systems | HAF012 | 1:1000 western blot |
| Software, algorithm | Prism | GraphPad | RRID:SCR_002798 | https://www.graphpad.com/features |
| Software, algorithm | Adobe Photoshop | Adobe Systems | RRID:SCR_014199 | https://www.adobe.com/ |
| Software, algorithm | Adobe Illustrator | Adobe Systems | RRID:SCR_010279 | https://www.adobe.com/ |
| Software, algorithm | FIJI/ImageJ | *Shih and Yamada, 2012* #370 | RRID:SCR_002285 | https://imagej.net/Fiji/Downloads |

