## [Editor Report]

Contact inhibition of locomotion (CIL) describes a process where collision between migrating cells induces a change in direction of cell migration leading to cell dispersion and Schwann cell migration has been shown to be essential for nerve repair. In this study, Hoving and colleagues demonstrate that CIL is an important driver of collective Schwann cell migration by using live cell imaging, cell migration assays and an ex vivo model of nerve injury and repair. They show that Slit2/3/Robo signaling is required to induce local cell repulsion between colliding Schwann cells, in an N-cadherin-dependent manner and further demonstrate the importance of an N-cadherin-glypican-4-Slit signaling axis during CIL. This study convincingly describes a new and important molecular mechanism of CIL in adult tissues and in the context of wound repair and is likely to be of interest to a broad range of biologists given the importance of both CIL and N-cadherin to a number of cellular and developmental processes.

---

## [Decision Letter]

**Decision letter after peer review:**

Thank you for submitting your article "N-cadherin directs the collective Schwann cell migration required for nerve regeneration through Slit2/3 mediated contact inhibition of locomotion" for consideration by *eLife*. Your article has been reviewed by 3 peer reviewers, one of whom is a member of our Board of Reviewing Editors, and the evaluation has been overseen byJonathan Cooper as the Senior Editor.

Essential revisions (for the authors):

1. Why does the Ncad/Ncad condition still have a high level of repulsion? Circa 60%? This is more than when looking at ctl+Ncad cells (below 40% repulsion Figure 2b). This does not seem logical. If none of the cells has Ncad to present slit how is repulsion triggered? According to the authors' model, cells should not be able to be repelled and not be able to form stable junctions. Can Glypican-4 present slit in absence of Ncad? Along these lines, each of the single knockdown (N-cad Figure 1g, slit Figure 4d, Gly Figure 4a) still leads to a relatively high rate of repulsion in cell collision assays, circa 40-50% compared to 60-70% in controls. This seems to suggest that these molecules may act in parallel and not only through one pathway. Even if they are all in the same pathway, the reduction of repulsion remains modest. That would mean that other signals may contribute to repulsion but this is not discussed.

2. In Figure 2, the authors score cell repulsion events between Schwann cells expressing siRNA against N-cadherin and control cells. The data in this figure show that cell repulsion occurs only when N-cadherin is present on one or both colliding cells. The graph in Figure 2b is not intuitive. Are the legends on the x-axis reflecting the cell which was tracked meeting a similar/different cell? E.g., N-Cad:Control implies N-cad cell was tracked until it meets a control cell etc and the bars reflect the cell response of the tracked cell. This is important to know to fully understand the data because the N-Cad:control versus Control:N-cad show different data, suggesting different responses depending on which cell was tracked. Otherwise, the data should be the same here to support the conclusion that N-cadherin presents a repulsive cue but is not required to respond to repulsive cue.

3. Following on from this, what happens to Robo/Slit2/3 localisation (and expression) at the site of collision? Can you see this by immunofluorescence with or without N-cadherin expression? You show very nicely that Slit2/3 protein localisation is dependent on N-cadherin; however, not at collision sites (also see comment below). The data suggest that N-cadherin is presenting Slit2/3 ligand at the membrane and subsequently to colliding cells. Alternatively, Slit2/3 and Robo localisation are both regulated by N-cadherin. One could speculate that presentation of ligand at the membrane is changing with respect to N-cadherin, whereas Robo levels/localisation remains unchanged, regardless of N-cadherin expression.

4. The authors show Slit2/3 localises in vesicles that move to the edge of cell in an N-cadherin-dependent manner. Western blots suggest protein expression is unchanged; however, immunofluorescence images suggest protein levels are also changing. Can you clarify on this? The descriptions used indicate the staining reflects cell surface protein; however, this would not match protein in vesicles. My interpretation is that Slit ligands are recycling in cells and trafficked to cell-cell contacts following N-cadherin-based cell-cell contacts (See comment 2 above). Do you see Slit ligands in endosomes/recycling endosomes and is this dependent on N-cadherin expression and/or engagement/collision of cells?

5. Glycipan-4 data is interesting. Is it possible that glycipan-4 is required for N-cadherin turnover/recycling? Do you see N-cadherin localisation/expression change in glycipan-4 depleted cells? Equally, glycipan-4 may regulate recycling of Slit2/3 ligands. Analysis of N-cadherin and Slit2/3 ligand expression/localisation in glycipan-4 depleted cells with/without cell collision would help here. The proposed physical cooperation between Glypican4 and N-cad is solely based on the proteomics analysis identifying Glypican-4 peptides following N-cad pull-down. If these two proteins act through the same pathway the double knockdown should be as strong as their single knockdown. Such double knockdown should be performed. In addition, an alternative method to study colocalization should be used to strengthen the physical interaction hypothesis. As I understand it the authors do not have a working antibody against Gly4. Is that Correct? If so, can an exogenous myc-tagged Glypican-4 be used to pull-down N-cad? Is a GFP-tagged Glypican-4 colocalized with the immunodetection of endogenous N-cad. The authors can decide and what is technically feasible in their system to address this point.

6. Authors describe how CIL propels cells to migrate as a collective and loss of Slit2/3/Robo signalling leads to a loss of cell repulsion. This conclusion is supported by the data; however the mechanism is missing. While I understand that mechanisms underpinning change in cell polarity would require additional experiments outside the scope of this study, I wondered whether the authors could reframe some of the analysis in terms of persistence of migrating cells, especially in the ex vivo model. One would expect cells migrating as cords to show persistence with directionality; whereas those treated with rSlit2 would lose this persistence. This could be analysed with regards to front/back polarity relative to the leading cell.

7. I could not find any controls for efficiency and/or specificity for the siRNAs against Robo1/2. Please provide such controls or refer to publication(s) that used the same siRNA in which proper controls were performed. In absence of controls, data from the use of these siRNA should be removed.

8. The statements about polarity under various experimental conditions should be confirmed by a proper analysis of cell polarity for instance using an anti-Rac1-GTP staining or a golgi staining and plotting the golgi/Nucleus axis or any other alternative that the authors find suitable. If not such statement should either be removed or significantly toned down. On example is the comparison between *Sox2* + rSlit2 and siRNA slit (Figure 6). In rSlit 2 contrary to what the authors say polarity seems to be present with area of flat membrane at the free edge suggesting that cells are still polarized according to cell-cell contact whereas in siSlit no such flat membranes are seen at the edge of cells.

9. Statistics. Authors have used t-test (or equivalent) to compare percentages across multiple experimental conditions. However, to my (limited) knowledge, t-tests are not designed to compare proportions. I think alternative methods need to be used or a precise argumentation as to why such tests can be applied here should be provided.

10. SiRNA Ncad seems to have no effect on slit expression and protein level on western blot but by IF the slit signal is almost lost. Is Ncad loss of function having an effect on slit protein levels or not? These two datasets are at odds with one another. Also these data are used to conclude that N-cad is required for Slit to be at the cell surface but neither of these techniques is designed to study that. If authors wish to study the Slit trafficking outside the cell or bound at the surface they need to use techniques such as Halo-tag and perform a pulse-chase assay with a short exposure of cells to a fluorescently tagged version of the Halo-tag and see what happens to slit2/3 in controls vs siN-cad conditions. Or any alternative method designed to monitor Slit at the cell surface (i.e. detection of slit in the enriched medium of control vs siNcad cells or immunodetection of slit without detergent to avoid entry of the anti-slit antibody inside the cells). Also it is not clear if the authors wish to say that N-cad allows slit to reach the membrane or to stay presented at the membrane.

---

## [Author Response]

Essential revisions (for the authors):1. Why does the Ncad/Ncad condition still have a high level of repulsion? Circa 60%? This is more than when looking at ctl+Ncad cells (below 40% repulsion Figure 2b). This does not seem logical. If none of the cells has Ncad to present slit how is repulsion triggered? According to the authors' model, cells should not be able to be repelled and not be able to form stable junctions. Can Glypican-4 present slit in absence of Ncad? Along these lines, each of the single knockdown (N-cad Figure 1g, slit Figure 4d, Gly Figure 4a) still leads to a relatively high rate of repulsion in cell collision assays, circa 40-50% compared to 60-70% in controls. This seems to suggest that these molecules may act in parallel and not only through one pathway. Even if they are all in the same pathway, the reduction of repulsion remains modest. That would mean that other signals may contribute to repulsion but this is not discussed.

This is an important point and is relevant to many of the concerns raised by the Reviewers. These concerns relate to the apparent residual repulsive activity in Ncadherin and Glypican-4 knock-down cells, which might reflect an additional repulsive signal is present. However, these results need to be considered in the overall context of how SCs migrate. If one watches the videos, it is apparent that even without any cell:cell contacts, SCs put out and retract multiple protrusions around the cell, and appear to change direction randomly. What this means is that there is a background level of retraction seen when the cells make contact. In earlier drafts of the manuscript, we included a more detailed description of this behaviour, which explained these caveats. We apologise for removing this section, as it is clear that this has led to confusion. Instead, the measurement which is more meaningful in this assay is the increase in “overlapping” or “no retraction” behaviours, which are seen rarely in control cells. For example, in the videos, it can clearly be observed that protrusions in N-cadherin knockdown cells, frequently cross over an encountered cell whereas this is rarely seen in control cells. Importantly, all N-cadherin knockdown cells show no apparent recognition of an encountered cell, in that they do not change direction but instead migrate over the surface of the encountered cell.

To help clarify the behaviour of these cells and that knockdown of N-cadherin eliminates repulsive behaviour between SCs, we have made three changes to improve the manuscript.

Provided more text to better explain repulsive SC behaviour and the nature of the assay (Page 5, Line 17).

Changed the nomenclature for this specific assay describing the behaviour of specific protrusions, so that (i) repulsion is changed to retraction (ii) adhesive to no retraction, and (iii) invasive to overlapping.

To further clarify this issue, we have performed a different quantification of the CIL assay, which determines whether the cell changes its direction of movement following an encounter. This measurement uses vector analysis to analyse interactions between migrating cells, by calculating the displacement of a migrating cell prior to (Vector A) and following a collision (Vector B) (Paddock and Dunn, 1986). The difference in displacement between the vectors (Vector B-A) represents the difference between how far the cell has progressed and how far it would have migrated had it not encountered another cell. Free moving cells which did not collide with other cells were also analysed over the same duration. Cells exhibiting CIL have a negative value indicating that the cell has changed direction following collision.

Using this assay, it can clearly be observed that the cellular repulsion signal is lost in Ncadherin K/D cells, with the movement of these cells after contact indistinguishable from free moving cells (new Figure 1h). In control cells, a significant difference between free moving and colliding cells is observed, as control cells show a change in direction following a collision. This is also clearly seen by observation of the videos (Video 1) and the representative still images in Figure 1f. These results more clearly demonstrate that there is no residual repulsive signal present in the N-cadherin knockdown cells.

2. In Figure 2, the authors score cell repulsion events between Schwann cells expressing siRNA against N-cadherin and control cells. The data in this figure show that cell repulsion occurs only when N-cadherin is present on one or both colliding cells. The graph in Figure 2b is not intuitive. Are the legends on the x-axis reflecting the cell which was tracked meeting a similar/different cell? E.g., N-Cad:Control implies N-cad cell was tracked until it meets a control cell etc and the bars reflect the cell response of the tracked cell.

Yes, this is the case, we tracked a cell from one group and then determined its response upon collision with the second cell type e.g. N-Cad:Control will measure the behaviour of a N-cadherin K/D cell following an encounter with a control cell.

This is important to know to fully understand the data because the N-Cad:control versus Control:N-cad show different data, suggesting different responses depending on which cell was tracked. Otherwise, the data should be the same here to support the conclusion that N-cadherin presents a repulsive cue but is not required to respond to repulsive cue.

We have modified the axis and legends (new Figures 2b and d), as we agree, it could be clearer. We have also included a cartoon for clarity (new Figure 2c).

3. Following on from this, what happens to Robo/Slit2/3 localisation (and expression) at the site of collision? Can you see this by immunofluorescence with or without N-cadherin expression?

Firstly, without N-cadherin, we do not observe Slit2/3 at contact sites as upon Ncadherin knockdown, we no longer observe Slit2/3 in the protrusions (Figure 5a,c). Secondly, it is hard to address the dynamics of Slit2/3 localisation at cell contacts prior to and after CIL in fixed cells and we currently do not have the technologies available to visualise Slit2/3 localisation by live-imaging. However, we do observe Slit2/3 at collision sites as well as on the membrane of cells not in contact, which might indicate that the localisation remains unchanged. We have included new Figures to show this is the case (new Supplementary Figure 5h).

You show very nicely that Slit2/3 protein localisation is dependent on N-cadherin; however, not at collision sites (also see comment below). The data suggest that N-cadherin is presenting Slit2/3 ligand at the membrane and subsequently to colliding cells. Alternatively, Slit2/3 and Robo localisation are both regulated by N-cadherin. One could speculate that presentation of ligand at the membrane is changing with respect to N-cadherin, whereas Robo levels/localisation remains unchanged, regardless of N-cadherin expression.

Using available antibodies, we have been unable to convincingly detect Robo by immunofluorescence. However, our mixing experiments showed that cells are still repulsed when N-cadherin is knocked-down, which would imply the Robo receptor is still capable of enacting a repulsion signal (Figure 2a-b). We have clarified this point in the manuscript (Page 17, Line 18).

4. The authors show Slit2/3 localises in vesicles that move to the edge of cell in an N-cadherin-dependent manner. Western blots suggest protein expression is unchanged; however, immunofluorescence images suggest protein levels are also changing. Can you clarify on this?

It is the case we show that following loss of N-cadherin, overall Slit2/3 protein and mRNA levels are unaltered as measured by RT-qPCR and western blot analysis (Supplementary Figure 5a-b). We also showed by immunostaining that N-cadherin KD results in a change in Slit2/3 localisation (Figure 5a-d), with Slit2/3 becoming restricted to the perinuclear area and no longer reaching the cell protrusions. We agree the original selected images were slightly misleading, as although they showed nicely that Slit2/3 is lost from protrusions in N-cadherin knockdown cells, the expression in the rest of the cell was not clearly visible. We have now changed and included new images to show more clearly the overall staining (Supplementary Figure 5g) and quantified the expression from immunostaining images, which also show no changes in overall expression levels (new Supplementary Figure 5e-f).

The descriptions used indicate the staining reflects cell surface protein; however, this would not match protein in vesicles.

It is well characterised that Slit2/3 function at the cell surface as repulsive molecules (Brose et al., 1999) (De Bellard et al., 2003). We also now include new images in which we can more clearly observe Slit2/3 at the cell surface (new Supplementary Figure 5h). It is also the case that Slit2/3 are detected within vesicles, which are presumably trafficking Slit2/3 to and from the cell-surface. This dynamic behaviour would be consistent with our findings that N-Cadherin is a highly dynamic protein, arriving in continual waves to the cell surface and junctions (Supplementary Figure 5d, Video 6), which would fit with the requirement for N-cadherin to traffic Slit2/3 to the plasma membrane.

My interpretation is that Slit ligands are recycling in cells and trafficked to cell-cell contacts following N-cadherin-based cell-cell contacts (See comment 2 above). Do you see Slit ligands in endosomes/recycling endosomes and is this dependent on N-cadherin expression and/or engagement/collision of cells?

We cannot rule this out this interpretation, but our observations do not detect an obvious difference in the levels of Slit2/3 at the surface of protrusions of cells in contact with other cells compared to the protrusions of freely-moving cells (new Supplementary Figure 5h). Further work needs to establish the exact mechanism for how N-cadherin is required to traffic Slit2/3 towards the protrusions but we believe this is beyond the scope of this paper. However, we have now added this speculation to the Discussion section (Page 17, Line 11).

5. Glycipan-4 data is interesting. Is it possible that glycipan-4 is required for N-cadherin turnover/recycling? Do you see N-cadherin localisation/expression change in glycipan-4 depleted cells? Equally, glycipan-4 may regulate recycling of Slit2/3 ligands. Analysis of N-cadherin and Slit2/3 ligand expression/localisation in glycipan-4 depleted cells with/without cell collision would help here.

We show that upon Glypican-4 KD, increased N-cadherin is observed at cell contacts. (Figure 4b). We interpreted this as a strengthening of the N-cadherin junctions over-time in the absence of a repulsive signal. It is possible that Glypican-4 could control Ncadherin turnover, however it should be noted that we see a similar increase in Ncadherin junctions when we deplete Slit2/3 or Robo1 and 2, which suggests ours is the correct interpretation.

We have now analysed the effects of Glypican-4 depletion on Slit2/3 expression and the K/D does not mimic N-cadherin KD in that we still observe Slit2/3 in vesicles within the protrusions (New Supplementary Figure 5i). This result is consistent with the suggested role for Glypicans in presenting the secreted Slit ligand on the cell-surface (Ronca et al., 2001). Further work would be required, with reagents which currently don’t exist, to address this possibility, which we think is beyond the scope of this paper.

The proposed physical cooperation between Glypican4 and N-cad is solely based on the proteomics analysis identifying Glypican-4 peptides following N-cad pull-down. If these two proteins act through the same pathway the double knockdown should be as strong as their single knockdown. Such double knockdown should be performed.

We have performed the double-knockdown and the phenotype mimics the N-cadherin KD as would be predicted by our model (Page 10, Line 9, New Supplementary Figures 4d-e).

In addition, an alternative method to study colocalization should be used to strengthen the physical interaction hypothesis. As I understand it the authors do not have a working antibody against Gly4. Is that Correct? If so, can an exogenous myc-tagged Glypican-4 be used to pull-down N-cad? Is a GFP-tagged Glypican-4 colocalized with the immunodetection of endogenous N-cad. The authors can decide and what is technically feasible in their system to address this point.

We have attempted pull-down experiments but could not get appropriate localisation of the tagged forms. We also tested several antibodies, none of which were convincing enough to use for a pull down or worked in western blot assays or immunostaining, using knockdown Glypican-4 cells as controls.

6. Authors describe how CIL propels cells to migrate as a collective and loss of Slit2/3/Robo signalling leads to a loss of cell repulsion. This conclusion is supported by the data; however the mechanism is missing. While I understand that mechanisms underpinning change in cell polarity would require additional experiments outside the scope of this study, I wondered whether the authors could reframe some of the analysis in terms of persistence of migrating cells, especially in the ex vivo model. One would expect cells migrating as cords to show persistence with directionality; whereas those treated with rSlit2 would lose this persistence. This could be analysed with regards to front/back polarity relative to the leading cell.

To measure persistent migration in the cords in the ex-vivo model, we have taken several approaches. Firstly, the original quantifications showed increased migration of the control cells compared to those treated with rSlit2 (Figure 7c-d). Secondly, we showed that the control SCs are found in cords, whereas those treated with rSlit2 are more commonly found in clusters (Figures7e-g). Together these results indicate persistent, collective migration of the cords that is lost in cells treated with rSlit2. However, to analyse persistent migration more directly, we have now done the following: (i) As suggested by the Reviewers, we have quantified the polarity of following cells relative to the leading cell using the nucleus, as an indicator of persistent, polarised migration (new Figure 7j and Supplementary 7d), (ii) We have also quantified the directionality of the SC groups and find a higher level of directionality of the control cells, as they migrate across the wound site, as predicted (New figure 7k).

7. I could not find any controls for efficiency and/or specificity for the siRNAs against Robo1/2. Please provide such controls or refer to publication(s) that used the same siRNA in which proper controls were performed. In absence of controls, data from the use of these siRNA should be removed.

These were omitted in error and have now been added to Supplementary Figure 4 (new Supplementary Figure 4h).

8. The statements about polarity under various experimental conditions should be confirmed by a proper analysis of cell polarity for instance using an anti-Rac1-GTP staining or a golgi staining and plotting the golgi/Nucleus axis or any other alternative that the authors find suitable. If not such statement should either be removed or significantly toned down. On example is the comparison between Sox2 + rSlit2 and siRNA slit (Figure 6).

We are unable to use conventional polarity analysis such as the Golgi/Nucleus axis, as we don’t observe that the Golgi aligns with the polarisation of the nucleus in control Schwann cells, when they migrate in a directional manner, as has also been found for other cell types (Uetrecht and Bear, 2009) (Pouthas et al., 2008) (Vaidžiulytė et al., 2022). However, as the nucleus polarises beautifully in migrating Schwann cells, we have instead quantified the alignment of the nuclei in each cluster to complement our original analysis (new Supplementary Figure 6d). As expected, in *Sox2* control cells, the nuclei are aligned in a manner indicative of the attempted outward migration of these cells, whereas the nuclei are more randomly aligned in the *Sox2* rSlit2-treated cells (new Figure 6i) and Slit2/3 KD cells (new Supplementary Figure 6g) consistent with the loss of the repulsion signal resulting in a failure of these cells to polarise.

In rSlit 2 contrary to what the authors say polarity seems to be present with area of flat membrane at the free edge suggesting that cells are still polarized according to cell-cell contact whereas in siSlit no such flat membranes are seen at the edge of cells.

We do still see some flat membrane at the free edges of the clusters in both rSlit2 and siSlit-treated cells, although it varies between the clusters. However, what is consistent between both groups is the decreased polarisation of the cells. This was measured in the original submission by comparing the roundness of the cells within the clusters. As discussed above, we have now extended this analysis to show the polarisation of the clusters by measuring the alignment of the cells within the clusters. To do this we have used the nuclear axis as a measure of each cells individual polarisation and compared this to the group (new Figures 6i and Supplementary Figure 6g) as well as quantifying the overall cluster shape (new Supplementary Figure 6e-f). These data clearly show, consistent with the videos, that Slits mediate the polarisation of the clusters.

9. Statistics. Authors have used t-test (or equivalent) to compare percentages across multiple experimental conditions. However, to my (limited) knowledge, t-tests are not designed to compare proportions. I think alternative methods need to be used or a precise argumentation as to why such tests can be applied here should be provided.

Thank you for bringing this to our attention. We have now presented the data as the total number of events in each condition rather than as a proportion and have performed the appropriate statistics for each experiment.

10. SiRNA Ncad seems to have no effect on slit expression and protein level on western blot but by IF the slit signal is almost lost. Is Ncad loss of function having an effect on slit protein levels or not? These two datasets are at odds with one another. Also these data are used to conclude that N-cad is required for Slit to be at the cell surface but neither of these techniques is designed to study that. If authors wish to study the Slit trafficking outside the cell or bound at the surface they need to use techniques such as Halo-tag and perform a pulse-chase assay with a short exposure of cells to a fluorescently tagged version of the Halo-tag and see what happens to slit2/3 in controls vs siN-cad conditions. Or any alternative method designed to monitor Slit at the cell surface (i.e. detection of slit in the enriched medium of control vs siNcad cells or immunodetection of slit without detergent to avoid entry of the anti-slit antibody inside the cells). Also it is not clear if the authors wish to say that N-cad allows slit to reach the membrane or to stay presented at the membrane.

As discussed earlier, we now provide further data, which shows that overall Slit2/3 expression as detected by IF is not decreased by N-cadherin K/D, consistent with our western blot analysis (new Supplementary Figure 5e-f). Instead, our findings show that Slit2/3 is no longer present in vesicles that emanate towards the plasma membrane in K/D cells. We also now show further images that clearly show Slit2/3 at the plasma membrane (new Supplementary Figure 5h) and that this is lost in N-cadherin K/D cells (new Supplementary Figure 5g). Extensive prior literature has shown that Slit2/3 signal from the cell-surface (Blockus and Chedotal, 2016) (De Bellard et al., 2003) (Brose et al., 1999) and all our results are consistent with these findings. We are therefore confident of the interpretation of our results.

We have considered the suggested experiments but, in our hands, tagged-proteins do not localise appropriately and even the lightest fixation treatment, permeabilises our primary cells. We currently cannot formally distinguish between whether N-cad allows Slit2/3 to reach the membrane or reflects a failure to stay presented. However, the lack of Slit2/3 in vesicles in N-cadherin K/D cells would suggest that it is not being delivered, as a failure to remain presented would indicate faster turnover, which would be expected to result in the continued presence of Slit2/3 in vesicles. Therefore, we are comfortable with our conclusions but have attempted to further clarify the text (Page 17, Line 11).